JCB Journal of Cell Biology

# Regulated targeting of the monotopic hairpin membrane protein Erg1 requires the GET pathway

Ákos Farkas[1], Henning Urlaub[2,3], Katherine E. Bohnsack[1], and Blanche Schwappach[1]

The guided entry of tail-anchored proteins (GET) pathway targets C-terminally anchored transmembrane proteins and protects cells from lipotoxicity. Here, we reveal perturbed ergosterol production in Δ*get3* cells and demonstrate the sensitivity of GET pathway mutants to the sterol synthesis inhibiting drug terbinafine. Our data uncover a key enzyme of sterol synthesis, the hairpin membrane protein squalene monooxygenase (Erg1), as a non-canonical GET pathway client, thus rationalizing the lipotoxicity phenotypes of GET pathway mutants. Get3 recognizes the hairpin targeting element of Erg1 via its classical client-binding pocket. Intriguingly, we find that the GET pathway is especially important for the acute upregulation of Erg1 induced by low sterol conditions. We further identify several other proteins anchored to the endoplasmic reticulum (ER) membrane exclusively via a hairpin as putative clients of the GET pathway. Our findings emphasize the necessity of dedicated targeting pathways for high-efficiency targeting of particular clients during dynamic cellular adaptation and highlight hairpin proteins as a potential novel class of GET clients.

## Introduction

Biological membranes are composed of lipids and proteins. The specific combination of lipids and membrane proteins endows different cellular membranes, such as the plasma membrane and intracellular membranes of eukaryotic cells, with appropriate functional properties (van Meer et al., 2008). A huge proportion of lipid and protein biosynthesis takes place at the ER, an organelle that connects to many other cellular compartments via the secretory pathway and via organelle contact sites (Bohnert, 2020; Olzmann and Carvalho, 2019; Scorrano et al., 2019; Wu et al., 2018). The interplay of protein biogenesis and lipid metabolism at the ER has emerged as an important determinant of the unfolded protein response (Stordeur et al., 2014; Surma et al., 2013). However, there is currently little insight into the role of protein targeting in the regulation of lipid metabolism.

The GET pathway, or TRC40 pathway in mammals, is a highly conserved targeting pathway that recognizes C-terminally anchored membrane proteins (tail-anchored [TA] proteins) after synthesis at the ribosome (Borgese et al., 2019; Farkas and Bohnsack, 2021; Mateja and Keenan, 2018; Shan, 2019). A cytosolic pre-targeting complex, composed of Sgt2 and the Get4-Get5 heterodimer in *Saccharomyces cerevisiae* (yeast), or SGTA, TRC35, UBL4A, and BAG6 in mammals, mediates the initial capture of nascent TA proteins (Jonikas et al., 2009; Kohl et al., 2011; Leznicki et al., 2013; Mariappan et al., 2010; Shao et al.,

2017; Zhang et al., 2021). They are subsequently handed over to the cytosolic ATPase Get3 (yeast)/TRC40 (mammals), which delivers them to an ER-bound GET receptor complex composed of Get1 and Get2 (yeast) or WRB and CAML (mammals) for membrane integration (McDowell et al., 2020; Schuldiner et al., 2008; Stefer et al., 2011; Wang et al., 2014). Several GET gene deletion mutants were identified in a screen for yeast mutants hypersensitive to unsaturated fatty acids, and the number of cytosolic lipid droplets (LDs) was also reduced in these mutants (Ruggles et al., 2014). In combination, the two phenotypes may indicate impaired sterol metabolism as steryl esters are a major component of LDs, and the esterification of toxic fatty acids to a sterol moiety is an important tolerance strategy in yeast (Korber et al., 2017). The sterol biosynthesis pathway is well-defined and involves one TA protein, squalene synthase (Erg9 in yeast, SQS in mammals). However, mammalian SQS is integrated into the ER independently of the GET pathway (Chitwood et al., 2018; Coy-Vergara et al., 2019). Evidence of a functional link between the GET targeting pathway and sterol production, therefore, remains elusive. Here, we address a putative defect of yeast GET pathway mutants in sterol homeostasis, decipher its relationship to the evolutionarily conserved membrane protein targeting function of the GET pathway, and propose hairpin proteins as a novel class of Get3 clients.

[1]Department of Molecular Biology, University Medical Center Göttingen, Göttingen, Germany; [2]Bioanalytic Mass Spectrometry, Max-Planck Institute for Biophysical Chemistry, Göttingen, Germany; [3]Bioanalytics, Institute of Clinical Chemistry, University Medical Center Göttingen, Göttingen, Germany.

Correspondence to Blanche Schwappach: blanche.schwappach@med.uni-goettingen.de; Katherine E. Bohnsack: katherine.bohnsack@med.uni-goettingen.de.



## Results

### Ergosterol synthesis is impaired in Δget3 mutants

A functional, genome-wide evaluation of liposensitive yeast strains identified Δget1, Δget2, and Δget3 mutants as substantially sensitive to the monounsaturated fatty acid palmitoleate (Ruggles et al., 2014). Secondary analysis of the liposensitive hits revealed reduced LD content in the get mutants, despite the fact that no general correlation between liposensitivity and reduced LD numbers was observed. This finding raised the possibility that altered sterol metabolism is the underlying cause of liposensitivity of the get strains. As decreased levels of the membrane lipid ergosterol result in sensitivity to the antifungal drug terbinafine (Petranyi et al., 1984), we tested the growth of a strain lacking the central GET pathway component Get3 (Δget3) on a medium containing terbinafine. In line with the hypothesis that sterol synthesis is impaired in Δget3 mutants, the strain was strongly terbinafine sensitive (Fig. 1 A). The esterification of fatty acids to form steryl esters (SEs) is a major detoxification mechanism. At the same time, SEs are one of two main lipid classes in the hydrophobic core of LDs. To explore precisely how the lack of the GET pathway affects the cellular lipidome, we determined the levels of 20 lipid species in WT and Δget3 strains grown in full (yeast extract, peptone, dextrose; YPD) and synthetic complete (SC) media (Figs. 1 B and S1 and Table S1), as media composition is known to affect the lipidome substantially (Klose et al., 2012). The fungal sterol ergosterol is the functional equivalent of mammalian cholesterol. Indeed, ergosterol ester (EE) levels, as well as those of the other main LD constituent, triacylglycerol (TAG) were strongly reduced in the Δget3 strain in both growth conditions tested (Fig. 1 B). This identifies the lipid classes responsible for the lowered LD numbers observed in the get deletion strains (Ruggles et al., 2014). Decreased sterol biosynthesis not only correlates with less storage of EEs and TAGs in LDs (Rajakumari et al., 2008) but also with reduced levels of inositolphosphoryl-ceramide (IPC) as shown in mutants severely impaired in ergosterol biosynthesis (Guan et al., 2009). Consistent with this, IPCs were also significantly reduced in Δget3 cells in both growth conditions (Fig. S1). The ergosterol synthesis pathway (Fig. 1 C) commences with the synthesis of isoprenoids from acetyl-CoA in a series of cytosolic reactions (Klug and Daum, 2014). This initial phase comprises cytosolic reactions but is controlled by the ER-resident and sterol-regulated enzyme β-hydroxy-β-methylglutaryl-CoA reductase (Hmg1/2), and produces the activated isoprenoid lipid farnesyl-pyrophosphate. Condensation of two farnesyl-pyrophosphate molecules to squalene, catalyzed by squalene synthase (Erg9), and the formation of squalene 2,3-epoxide, catalyzed by squalene monooxygenase (Erg1), are the rate-limiting reactions of sterol synthesis. These and all subsequent steps leading to ergosterol occur in or at the hydrophobic environment of the ER membrane. Esterification and storage of ergosterol in LDs can provide cells with this essential membrane lipid independently of energy-consuming sterol synthesis.

Considering the role of the GET pathway in protein biogenesis, we hypothesized that the terbinafine sensitivity of the Δget3 mutant may reflect Get3-dependence of one or several of the enzymes involved in sterol biosynthesis (Fig. 1 C). Using a library of N-terminally GFP-tagged proteins expressed under the control of the moderately strong, constitutive NOP1 promoter (Weill et al., 2018) and a set of C-terminally GFP-tagged proteins expressed under the control of their endogenous promoters by individual genomic tagging (Sheff and Thorn, 2004), we tested all but one of the 30 enzymes collectively involved in sterol synthesis, storage, or mobilization for changes in their cellular distribution upon loss of Get3 (Figs. 1 D and S2). Erg20 could not be tagged either N- or C-terminally with GFP and was thus excluded from the analysis. Strikingly, Erg1, which is the direct target of terbinafine, displayed a distinguishable signal pattern with more diffuse background and less pronounced perinuclear ER in the Δget3 strain compared to WT. In contrast, localization of the only TA protein involved in ergosterol synthesis, Erg9, was not affected by the lack of Get3 (Fig. 1 D). This is in line with the observation that its conserved human homolog does not use the GET pathway for insertion into the ER membrane (Chitwood et al., 2018). Importantly, GFP–Erg9 and GFP-tagged Hmg1, Hmg2, Erg11, Erg24, Erg26, Erg28, Erg5, Erg4, Are1, Are2, and Atf2 (Fig. S2) showed robust fluorescent labeling of the ER in the Δget3 strain, which illustrates the intactness of the ER membrane and indicates that mislocalization of Erg1 to the cytosol in the Δget3 strain is not due to a general loss of proteins from the ER. Furthermore, the LD targeting of LD-localized proteins involved in ergosterol metabolism, including Erg7, Erg27, Erg6, Erg2, Tlg1, and Say1, appeared to be unaffected in the Δget3 strain as well (Fig. S2). Inducing a pulse of newly expressed TA protein is an established strategy to verify targeting pathway dependence (Schuldiner et al., 2008). Thus, to follow up on our initial observation that correct the localization of Erg1 to the ER requires Get3 (Fig. 1 D) and to test whether the mislocalization is due to defective biogenesis, we transiently overexpressed GFP–Erg1 from a galactose-inducible plasmid (Fig. 1 E). As expected for a biogenetic client of Get3, the acute induction of Erg1 expression from the GAL1 promoter exacerbated the phenotype substantially and rendered the faint ER pattern almost invisible against a diffuse cytosolic staining. As Erg1 is the direct target of terbinafine, we exploited the terbinafine sensitivity of the Δget3 strain to test whether strongly overexpressed GFP–Erg1 was able to rescue drug sensitivity (Fig. 1 F). The manipulation markedly increased the terbinafine tolerance of the WT strain, confirming the functionality of GFP–Erg1 and an increased capacity for converting squalene into squalene epoxide in this scenario. However, the Δget3 strain was equally sensitive to terbinafine irrespective of GFP–Erg1 induction, suggesting that, unlike in the WT, increased GFP–Erg1 expression does not lead to a substantially increased pool of functional GFP–Erg1 protein in Δget3 cells. Together, these data indicate that the efficient localization and function of Erg1 requires Get3.

### Squalene monooxygenase Erg1 is a client of the GET pathway

As Get3 has a broad client spectrum and Erg1 lacks a classic TA sequence, the dependence of the ER localization of Erg1 on Get3 could potentially be an indirect effect. To explore the possibility that Erg1 is a GET pathway client, potential direct interactions between these proteins were examined using the "client trap" variant of Get3 (Coy-Vergara et al., 2019; Powis et al., 2013), Get3

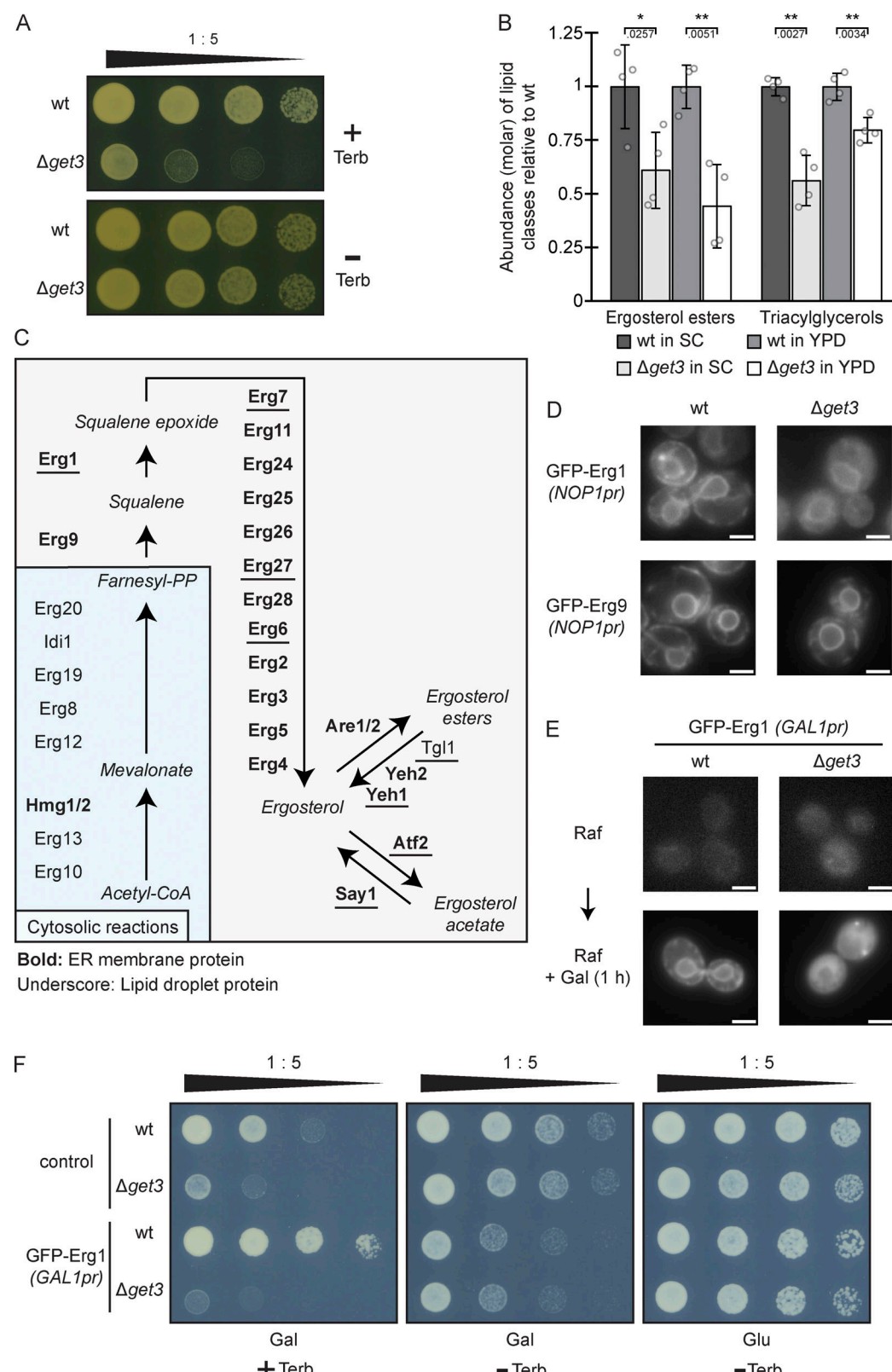

Figure 1. **The rate-limiting sterol biosynthesis enzyme Erg1 mislocalizes from the ER to the cytosol in Δ*get3* cells. (A)** Plate growth assay of WT and Δ*get3* strains in the presence and absence of terbinafine. Strains were spotted out in a one-to-five dilution series on YPD plates with (+ Terb) or without (− Terb) 50 µg/ml terbinafine and grown at 30°C. The images are representative of three biological replicates. **(B)** Lipidomic analysis of WT and Δ*get3* cells in both synthetic complete (SC) and full (yeast extract-peptone-dextrose; YPD) media is shown for ergosterol esters and triacylglycerols. Bars represent the average molar abundance of the indicated lipid classes from four biological replicates, with individual data points shown as gray dots, normalized to the WT strain in the respective media. Error bars indicate standard deviation of the mean. The P values calculated using the two-sided Welch's *t* test are shown with

numbers and represented as follows: * < 0.05; ** < 0.01. The remaining lipid classes are shown in Fig. S1. **(C)** Schematic representation of the ergosterol biosynthesis pathway in yeast. **(D)** Fluorescence microscopy images of WT and Δ*get3* strains expressing genomically N-terminally GFP-tagged Erg1 or Erg9 under the control of the *NOP1* promoter (pr). Images are representative of three biological replicates with >100 cells imaged for each replicate. Scale bar: 2 μM. **(E)** Fluorescence microscopy images of WT and Δ*get3* strains expressing *GAL1* promoter-driven N-terminally GFP-tagged Erg1 ectopically in synthetic dropout media containing 2% raffinose (Raf) or 1 h after transition to media containing 2% galactose (Gal). Images are representative of three biological replicates with >200 cells imaged for each replicate. Scale bar: 2 μM. **(F)** Plate growth assay of WT and Δ*get3* strains either expressing *GAL1* promoter-driven N-terminally GFP-tagged Erg1 ectopically or containing the control plasmid. Strains were spotted out in a one-to-five dilution series on synthetic dropout plates with (+ Terb) or without (− Terb) 50 μg/ml terbinafine (Terb), containing glucose (Glu) or galactose (Gal) as indicated.

D57E (DE), in which the ATPase activity of Get3 is impaired, preventing hand-over of clients to the GET receptor complex. Immunoprecipitation experiments were performed in the absence of detergent to avoid disrupting Get3-client interactions, and Get3 DE and associated proteins were eluted from a GFP affinity matrix using TEV protease to ensure specific elution. As a control for client binding to the hydrophobic groove of Get3 DE, we employed a variant Get3 D57E F190D I193D (DE FIDD), in which negatively charged side chains disturb the hydrophobic cavity formed by the Get3 dimer, thereby impeding interactions with clients that use this feature for binding (Coy-Vergara et al., 2019; Mateja et al., 2009). Being able to avoid detergent in the affinity purification protocol, we equally purified both forms of Get3 (Figs. 2 A and S3 and Table S2). We considered co-purifying proteins as enriched with either Get3 variant based on statistical significance (P < 0.05) and fold enrichment (>8, i.e., >3$_{log2}$). Pretargeting complex components Get4, Get5, and Sgt2, which more stably interact with Get3 in the presence of clients (Rome et al., 2014), were found in high abundance in both Get3 DE and Get3 DE FIDD samples, albeit Get5 preferentially enriched with Get3 DE compared with Get3 DE FIDD. As anticipated, Get3 DE enriched the well-established client protein Sed5 (Rivera-Monroy et al., 2016; Schuldiner et al., 2008) more strongly than Get3 DE FIDD. Strikingly, similar to Sed5, Erg1 was strongly enriched in both eluates but preferentially associated with Get3 DE compared with Get3 DE FIDD, supporting the notion that Erg1 is a direct client of Get3 that contacts the ATPase via its hydrophobic client-binding groove. To validate our mass spectrometry results, we sought to recapitulate the enrichment of Erg1 with Get3 DE via immunoblotting by using an N-terminally HA-tagged form of Erg1 expressed from its endogenous promoter (Fig. 2 B). Indeed, similarly to the well-characterized client TA protein Sed5, HA–Erg1 showed a fivefold higher enrichment with Get3 DE, as compared with Get3 DE FIDD (Fig. 2 C), and thus behaved as a bona fide Get3 client.

Get3 acts as a central component of the GET pathway, which comprises a cytosolic phase of client capture by Get4, Get5, and Sgt2 (Jonikas et al., 2009; Zhang et al., 2021), and receptor-dependent release of clients at the ER membrane involving a Get1/2 heterotetramer (McDowell et al., 2020; Schuldiner et al., 2008; Stefer et al., 2011; Wang et al., 2014). To address whether Get3-dependent targeting of Erg1 to the ER reflects a GET pathway-independent function of Get3 or if other steps of the GET pathway are required for its proper localization, the distribution of GFP–Erg1 in Δ*get5* and Δ*get1*Δ*get2* strains was monitored (Fig. 2 D). Erg1 mislocalized from the ER to the cytosol in both these strains, albeit only minimally in the absence of Get5, indicating that Erg1 targeting to the ER requires the GET

pathway. Compared with Δ*get3*, the weaker and stronger mislocalization observed in the Δ*get5* and Δ*get1*Δ*get2* strains, respectively, are in line with previous observations that the loss of the pretargeting complex components Get5 and Get4 causes a milder disruption of TA protein targeting than the lack of Get3 or Get1/Get2 (Jonikas et al., 2009). Further supporting the role of the GET pathway in targeting Erg1 to the ER, Δ*get5* and Δ*get1*Δ*get2* strains, like Δ*get3*, also displayed terbinafine sensitivity (Fig. 2 E) in line with the degree of Erg1 mislocalization. The strong defects observed in the Δ*get1*Δ*get2* strain may be rationalized by the fact that the GET pathway clients localize to cytosolic aggregates containing Get3 and pretargeting complex components in the absence of Get1 and Get2 (Jonikas et al., 2009; Powis et al., 2013; Schuldiner et al., 2008). This may exacerbate targeting defects by trapping clients in aggregates, thus making them inaccessible to alternative targeting pathways (Schuldiner et al., 2008). Together, these data indicate that the hairpin protein Erg1 is a GET pathway targeting client.

## Get3 targeting of Erg1 is mediated by its membrane-interacting hairpin

In contrast to the canonical TA protein clients of the GET pathway characterized by a single transmembrane segment (TMS) within 30 amino acids of their C-terminus (Schuldiner et al., 2008), Erg1 features two hydrophobic stretches at its C-terminus, analogous to its evolutionarily conserved mammalian homolog SQLE (Fig. S4, A and B). Compared with the TMS of the TA protein Sed5, both of these helices have a lower hydrophobicity due to the presence of more hydrophilic residues interrupting shorter stretches of hydrophobic residues (Fig. S4 B). The C-terminal region has long been suspected to be required for the membrane association of Erg1 (Leber et al., 1998), but the precise topology of the protein has not been experimentally determined. A recent structural model of SQLE (Padyana et al., 2019) revealed that the predicted C-terminal hydrophobic helices partially contact other parts of the protein, including the client-binding pocket of the enzyme (Fig. 3, A and B). However, they also remain partially accessible, potentially enabling them to anchor the protein into membranes by partial immersion (Allen et al., 2019). This structural feature is thus consistent with a monotopic topology that restricts SQLE, and likely yeast Erg1, to one face of the ER while being partially immersed into the membrane as an integral membrane protein. In line with a monotopic topology, Erg1 has been found to localize not only to the ER but also to LDs (Leber et al., 1998), which we confirmed by live-cell imaging using N- and C-terminal GFP tagged Erg1 and the lipid droplet stain monodansylpentane (MDH, Fig. 3 C). As LDs are surrounded by a lipid monolayer, the LD localization

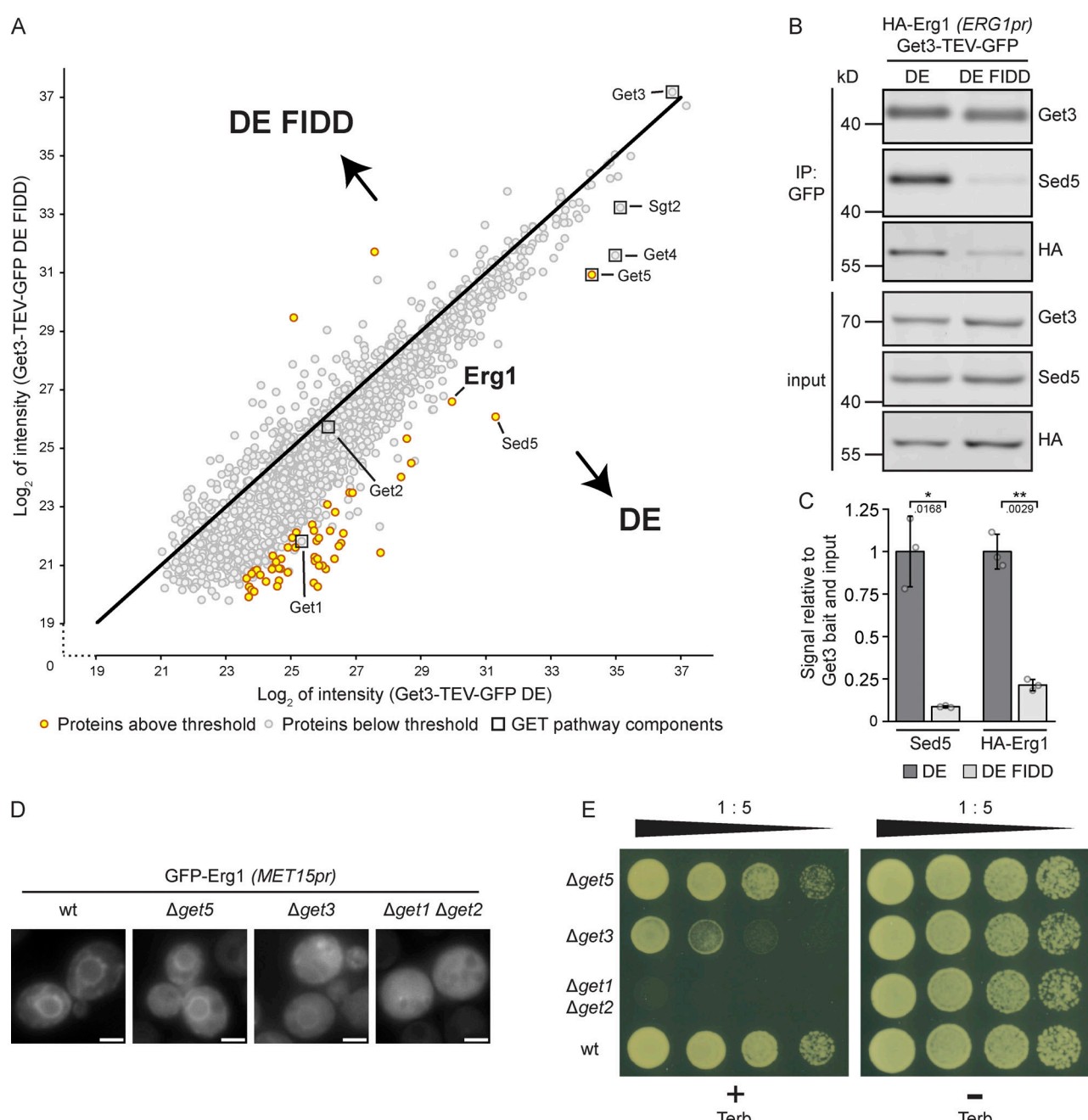

Figure 2. **Erg1 is a client of the GET pathway. (A)** Get3 D57E-TEV-GFP (DE) and Get3 D57E F190D I193D-TEV-GFP (DE FIDD) immunoprecipitates were analyzed by mass spectrometry. Immunoprecipitation was done in the absence of detergents. Axes represent the average $\log_2$ intensity of identified proteins in three biological replicates. Proteins with a more than eightfold enrichment and a statistical significance P < 0.05 (see Fig. S3 and Table S3) are marked in yellow and GET pathway components are indicated with boxes. The P values were calculated using the two-sided Welch's t test. **(B)** Immunoprecipitation performed using lysates from cells expressing HA–Erg1 and Get3 DE–TEV–GFP or Get3 DE FIDD–TEV–GFP. Immunoblotting of input and eluate (IP: GFP) samples was performed using antibodies against the proteins or tag indicated to the right of the panel. Lines mark images from the same membrane. Images are representative of three biological replicates. **(C)** Quantification of B. The signal corresponding to the indicated protein was normalized to Get3 in the eluate and the same protein in the corresponding input. Bars represent the average of three biological replicates with individual data points shown as gray dots. Error bars indicate standard deviation of the mean. The P values calculated using the two-sided Welch's t test are shown with numbers and represented as follows: * < 0.05; ** < 0.01. **(D)** Fluorescence microscopy images of cells lacking the indicated components of the GET pathway expressing N-terminally GFP-tagged Erg1 ectopically under the control of the *MET15* promoter (*MET15pr*). Images are representative of three biological replicates with >200 cells imaged for each replicate. Scale bar: 2 μM. **(E)** Plate growth assay of strains lacking the indicated components of the GET pathway in the presence and absence of terbinafine. Strains were spotted out in a one-to-five dilution series on YPD plates with (+ Terb) or without (– Terb) 50 μg/ml terbinafine and grown at 30°C. The images are representative of three biological replicates. Source data are available for this figure: SourceData F2.

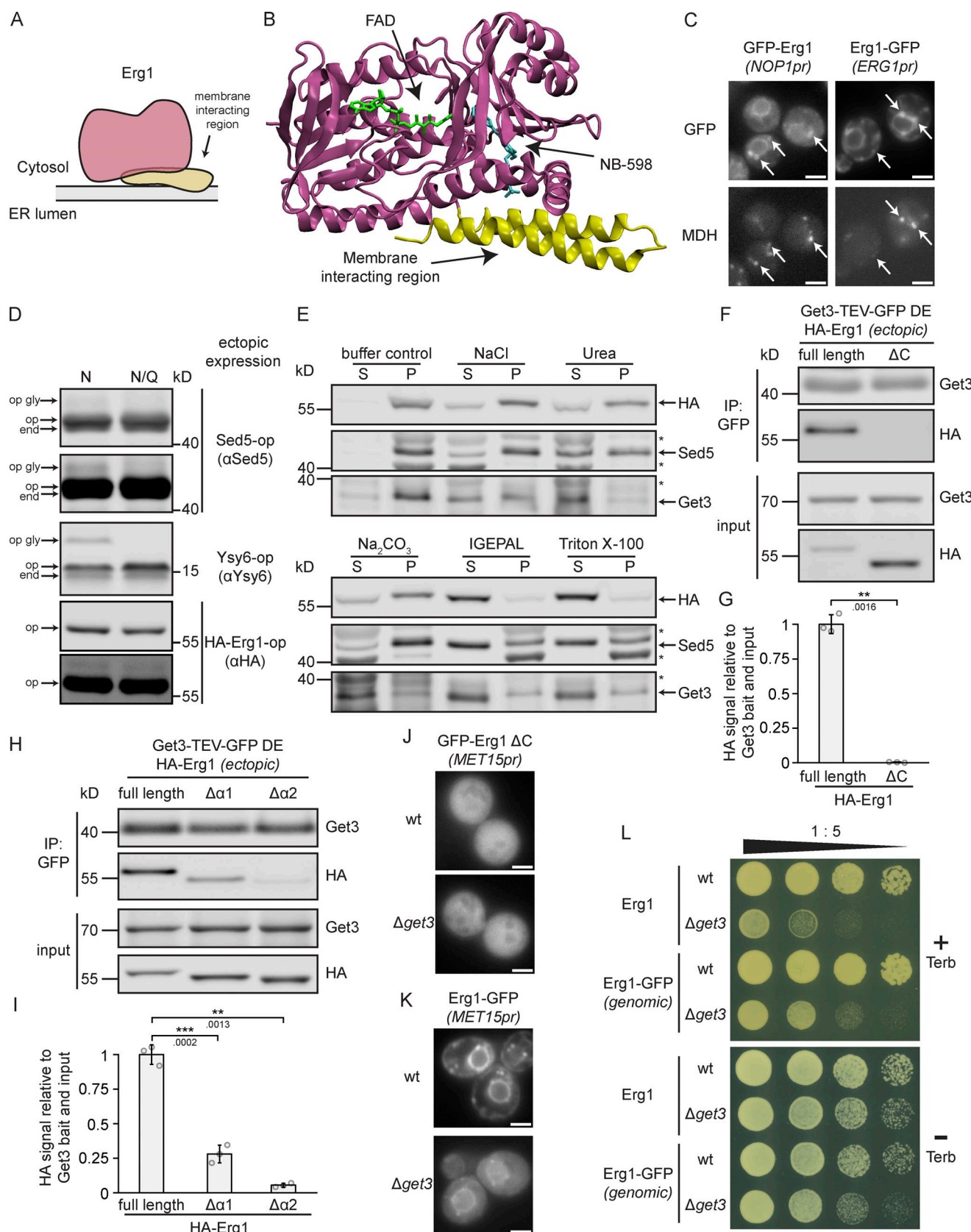

Figure 3. **Erg1 is a hairpin protein that can assume monotopic topology and associates with Get3 via its membrane-interacting region. (A)** Schematic representation of the topology of Erg1. **(B)** Structure of the human ortholog of Erg1, SQLE in complex with flavin adenine dinucleotide (FAD) and the small molecule inhibitor NB-589 that blocks the catalytic site (PDB accession no. 6C6P). The enzymatic core is shown in purple, the membrane-interacting helices in

yellow, and the N-terminal non-conserved region is missing. **(C)** Fluorescence microscopy images of strains expressing genomically C-terminally (Erg1–GFP) or N-terminally (GFP–Erg1) GFP-tagged Erg1. Cells were stained with the lipid droplet marker MDH. Images are representative of three biological replicates with >100 cells imaged for each replicate. Arrows indicate co-localization. Scale bar: 2 µM. **(D)** Immunoblot of cell lysates from a WT strain expressing the indicated constructs ectopically. The constructs contained either a standard glycosylatable opsin tag (N) or a non-glycosylatable form of it (N/Q). Antibodies used for detection are shown in brackets. Immunoblots of Sed5-op and HA-Erg1–op are shown with two different brightness settings. end: endogenously expressed protein; op: ectopically expressed non-glycosylated protein; op gly: ectopically expressed glycosylated protein. Images are representative of three biological replicates. **(E)** Immunoblot of microsomal protein from a strain expressing HA–Erg1 genomically. Isolated microsomes were incubated with storage buffer (buffer control), 0.5 M NaCl, 2.5 M urea, 0.1 M Na2CO3, 1% IGEPAL, or 1% Triton X-100 followed by ultracentrifugation. The immunoblot shows the proteins in the resulting supernatant (S) and pellet (P). Stars mark non-specific bands. The images are representative of three biological replicates. **(F)** Immunoblot of input and eluate of Get3 DE-TEV-GFP immunoprecipitates using antibodies against the proteins or tag indicated to the right of the panel. Lysates originated from Δget3 cells ectopically expressing Get3 DE-TEV-GFP and either full-length, N-terminally HA-tagged Erg1 (full length) or a truncated version lacking the C-terminal membrane interacting region (ΔC). Lines mark images from the same membrane. Images are representative of three biological replicates. **(G)** Quantification of F. Signal corresponding to indicated proteins was normalized to Get3 in the eluate and the same protein in the corresponding input. Bars represent the average of three biological replicates with individual data points shown as gray dots. Error bars indicate standard deviation of the mean. The P values calculated using the two-sided Welch's $t$ test are shown with numbers and represented as follows: ** < 0.01. **(H)** Immunoblot of input and eluate of Get3 DE-TEV-GFP immunoprecipitates using antibodies against the proteins or tag indicated to the right of the panel. Lysates originated from Δget3 cells ectopically expressing Get3 DE-TEV-GFP and either full-length, N-terminally HA-tagged Erg1 (full length) or a truncated version lacking the first (Δα1) or the second (Δα2) predicted helix of the C-terminal hairpin. Lines mark images from the same membrane. Images are representative of three biological replicates. **(I)** Quantification of H. Signal corresponding to indicated proteins was normalized to Get3 in the eluate and the same protein in the corresponding input. Bars represent the average of three biological replicates with individual data points shown as gray dots. Error bars indicate standard deviation of the mean. The P values calculated using the two-sided Welch's $t$ test are shown with numbers and represented as follows: ** < 0.01; *** < 0.001. **(J)** Fluorescence microscopy images of WT and Δget3 strains expressing N-terminally GFP-tagged Erg1 ΔC ectopically under control of the MET15 promoter (MET15pr). Images are representative of three biological replicates with >100 cells imaged for each replicate. Scale bar: 2 µM. **(K)** Fluorescence microscopy images of WT and Δget3 strains expressing C-terminally GFP-tagged Erg1 ectopically under control of the MET15 promoter (MET15pr). Images are representative of three biological replicates with >100 cells imaged for each replicate. Scale bar: 2 µM. **(L)** Plate growth assay of WT and Δget3 strains expressing either Erg1 or C-terminally GFP-tagged Erg1 (Erg1-GFP) from the endogenous ERG1 locus in the presence and absence of terbinafine. Strains were spotted out in a one-to-five dilution series on YPD plates with (+ Terb) or without (– Terb) 50 µg/ml terbinafine and grown at 30°C. The images are representative of three biological replicates. Source data are available for this figure: SourceData F3.

of Erg1 demonstrates that it can assume a monotopic conformation, although it cannot be excluded that Erg1 may also assume an alternative transmembrane topology with the complete penetration of the lipid bilayer at the ER membrane with the second, more hydrophobic helix of its hairpin serving as a TMS. To test this possibility, we fused a 13 amino acid long opsin-tag derived from bovine rhodopsin containing an N-glycosylation site (asterisk; GPNFYVPFSN*KTG) to the C-terminus of Erg1 and the TA proteins Sed5 and Ysy6, as has been reported before by Schuldiner et al. (2008). When exposed to the ER lumen, the opsin-tag is N-glycosylated, resulting in a visible electrophoretic shift upward during denaturing polyacrylamide gel electrophoresis. Control constructs were generated where the glycosylation site was replaced by a glutamine (N/Q), enabling definitive identification of bands corresponding to the glycosylated protein. As expected, both TA protein constructs showed some degree of glycosylation, with the ER-resident TA protein Ysy6-opsin being more glycosylated than the Golgi-resident Sed5-opsin (Figs. 3 D and S4 C). However, no glycosylation could be detected for the HA–Erg1–opsin construct (Fig. 3 D), supporting the notion that Erg1 does not expose its C-terminus to the ER lumen and hence does not assume TA protein topology.

The mammalian Erg1 homolog SQLE behaves partially as a peripheral membrane protein based on its extractability by alkaline pH (Coates et al., 2021). However, it is also possible for hairpin proteins to associate more tightly with membranes and to require detergent for efficient solubilization, as has been demonstrated for the yeast hairpin protein Tsc10 (Gupta et al., 2009). Indeed, HA–Erg1 was only partially solubilized by high salt concentration (0.5 M NaCl), denaturing conditions (2.5 M urea), or alkaline pH (0.1 M NA2CO3), which were sufficient to

extract Get3, which is peripherally associated with the ER membrane via its receptor (Fig. 3 E). HA–Erg1, like the TA protein Sed5, was only efficiently solubilized upon the addition of detergents (1% IGEPAL or 1% Triton X-100), providing further evidence in favor of a monotopic integral membrane topology (Fig. 3 E).

To test whether the C-terminal hydrophobic helices of Erg1 mediate binding to Get3, the co-enrichment of full-length HA-tagged Erg1 or a C-terminally truncated version lacking the hydrophobic helices (ΔC) with Get3 was analyzed. As expected for a Get3 client, full length, but not truncated, Erg1 was recovered (Fig. 3, F and G). To further dissect which of the helices of its hairpin mediate the interaction with Get3, we also tested the co-enrichment of HA–Erg1 lacking the first (Δα1, amino acids 441–463 deleted) or the second (Δα2, amino acids 464–498 deleted) hydrophobic helix with Get3 (Fig. 3, H and I). Interestingly, although both helices contribute significantly to Get3 binding, deletion of the first helix reduced Get3 binding by ~75%, whereas loss of the second helix, which is more hydrophobic than the first (Fig. S4 B), reduced Get3 binding by 95%.

Consistent with the hydrophobic helices functioning as a membrane anchor, Erg1 ΔC did not localize to the ER, but was present diffusely in the cytosol. Loss of Get3 had no visible effect on its localization (Fig. 3 J). Furthermore, we tested whether a construct in which Erg1 is targeted to the ER by replacing its C-terminal hairpin with the TMS of the TA protein Erg9 (Fig. S4 D) can functionally complement endogenous Erg1. To this end, we created a Δerg1 deletion mutant in the background of a URA3-carrying plasmid driving the expression of Erg1 tagged with a fluorescent protein. Strains were transformed with a

second plasmid driving the expression of GFP-tagged Erg1 lacking its C-terminal hairpin (ΔC) or having it replaced by the TMS of Erg9 (ΔC Erg9TMS). A plasmid expressing full-length Erg1 served as the positive and the empty vector as the negative controls. Counterselection of the *URA3*-containing Erg1 expression plasmid on a 5′-fluoroorotic acid (5-FOA) containing plate revealed that both Erg1 ΔC and Erg1 ΔC Erg9TMS failed to functionally rescue *Δerg1* (Fig. S4 E). This highlights that not only the localization of Erg1 to the ER but also the specific topological arrangement provided by its C-terminal hairpin is essential for the proper functioning of Erg1.

The position of the hydrophobic targeting segment with respect to a protein's termini is an important factor in determining targeting pathway dependence during membrane protein biogenesis. In fact, manipulating its relative position within the protein can divert clients to other pathways as has been shown for some Get3 clients (Aviram et al., 2016). Therefore, to test whether the C-terminal position of the hairpin of Erg1 drives its Get3 dependence, Erg1 was C-terminally GFP tagged so that the hairpin was in the middle portion of the protein. Surprisingly, ectopically expressed, C-terminally GFP-tagged Erg1 showed strong mislocalization to the cytosol in *Δget3* cells (Fig. 3 K). In line with this, C-terminally GFP tagged Erg1 expressed from its endogenous locus did not abolish the terbinafine sensitivity of *Δget3* cells (Fig. 3 L). This result suggests that, in contrast to a TA, the hydrophobic targeting element of Erg1 is recognized by Get3 in a binding mode that is not dependent on the C-terminal position of the hairpin.

**Targeting by Get3 enables a biogenetic pulse of Erg1 in response to terbinafine**

Like the *SED5* gene encoding the well-characterized Get3 TA client, Sed5, the *ERG1* gene is essential, whereas *GET3* is not. This implies that the action of other targeting pathways or chaperones ensure sufficient delivery of Get3 clients to allow survival in the absence of a functional GET pathway. However, high-fidelity targeting of Get3 clients to the ER membrane seems to be essential under stress conditions as evidenced by the fact that in the presence of terbinafine, increased Erg1 protein expression fails to rescue the drug sensitivity of the *Δget3* strain (Fig. 1 F). This implies that the kinetics or membrane specificity of Erg1 targeting are affected by the lack of Get3. To be able to assess the localization of Erg1 at its endogenous expression level after terbinafine treatment, we used strains expressing Erg1 tagged with GFP at its C-terminus from its endogenous locus. In line with previous reports (Leber et al., 1998), live-cell imaging of Erg1–GFP expressed from its native promoter indicated an increased Erg1 protein level in the presence of terbinafine (Fig. 4 A). To quantitatively assess the mislocalization of Erg1 in *Δget3* cells, we measured the average skewness of the distribution of the GFP signal in the cells. As skewness measures whether the distribution of pixel intensities of the image is skewed toward higher (positive skewness) or lower (negative skewness) values compared to the mean, the presence of bright pixels in the cells corresponding to ER and lipid droplet localization of Erg1 is reflected in a higher positive skewness as opposed to diffuse cytosolic staining. Indeed, a clear difference could be measured in

the distribution of the Erg1–GFP signal between logarithmically growing WT and *Δget3* cells (Fig. 4 B), which did not change after the addition of methanol (MeOH), which was used as a solvent for terbinafine. The difference between WT and *Δget3* cells became even more pronounced after the addition of terbinafine, as the skewness of the GFP signal became higher in WT cells due to the increasingly bright ER and lipid droplet signal, whereas the skewness of *Δget3* cells remained low due to the mislocalization of Erg1–GFP to the cytosol (Fig. 4, A and B).

To exclude the possibility that the GFP tag affected Erg1 stability, HA-tagged Erg1 expressed from its endogenous locus was used to accurately assess changes in protein levels after terbinafine treatment. Western blot analysis confirmed that Erg1 steady-state protein levels increased significantly in both the WT and the *Δget3* strains upon terbinafine treatment. Erg1 protein levels were lower in *Δget3* than in WT, which was already evident before terbinafine treatment, albeit to a smaller extent (Fig. 4, C and D). To determine whether the lower Erg1 protein level in the *Δget3* strain treated with terbinafine reflects decreased protein synthesis or decreased *ERG1* mRNA expression, we used reverse transcription quantitative polymerase chain reaction (RT-qPCR) to monitor *ERG1* mRNA levels. This confirmed a comparable increase of mRNA production from the *ERG1* gene in the presence of terbinafine in both WT and *Δget3* strains (Fig. 4 E), indicating that posttranslational events likely lead to lower Erg1 levels in *Δget3* cells. The increased *ERG1* mRNA expression we observed is likely due to the fact that impaired sterol biosynthesis reduces ergosterol levels and elicits transcriptional activation of genes bearing sterol-responsive elements in their promoter region, such as *ERG1* (Leber et al., 1998). In the absence of terbinafine, the expression of Erg1 from either its native promoter or the *NOP1* promoter supported cellular growth equally (Fig. 4 F). However, consistent with the notion that an upregulation of *ERG1* expression is required to withstand exposure to terbinafine and that the native *ERG1* promoter contains activation elements necessary for this regulated response, exchanging the native promoter for the *NOP1* promoter resulted in terbinafine sensitivity of an otherwise WT strain (Fig. 4 F). Taken together, these data support a model where terbinafine treatment induces a pulse of Erg1 synthesis essential for survival, which requires Get3 for high-fidelity and efficient targeting to the ER. However, inefficient targeting in the absence of Get3 leads to the degradation of mislocalized, non-ER-bound protein, reducing the amount of Erg1 at its site of function and hence leading to terbinafine sensitivity.

**Get3 affects Erg1 independently of the ubiquitin ligase Doa10**

Flexible adaptation of energy-demanding de novo sterol synthesis requires the regulated degradation of the participating enzymes. This is particularly relevant for Erg1 as it controls the flux from isoprenoids that have additional functions in metabolism to the pathway branch dedicated to producing sterols (Fig. 1 C). The ubiquitin ligase Doa10 is the major determinant of Erg1 turnover (Fig. 5 A), which is a process regulated by cellular sterol levels (Foresti et al., 2013). As Get3-dependent biogenesis and Doa10-mediated degradation both contribute to determining

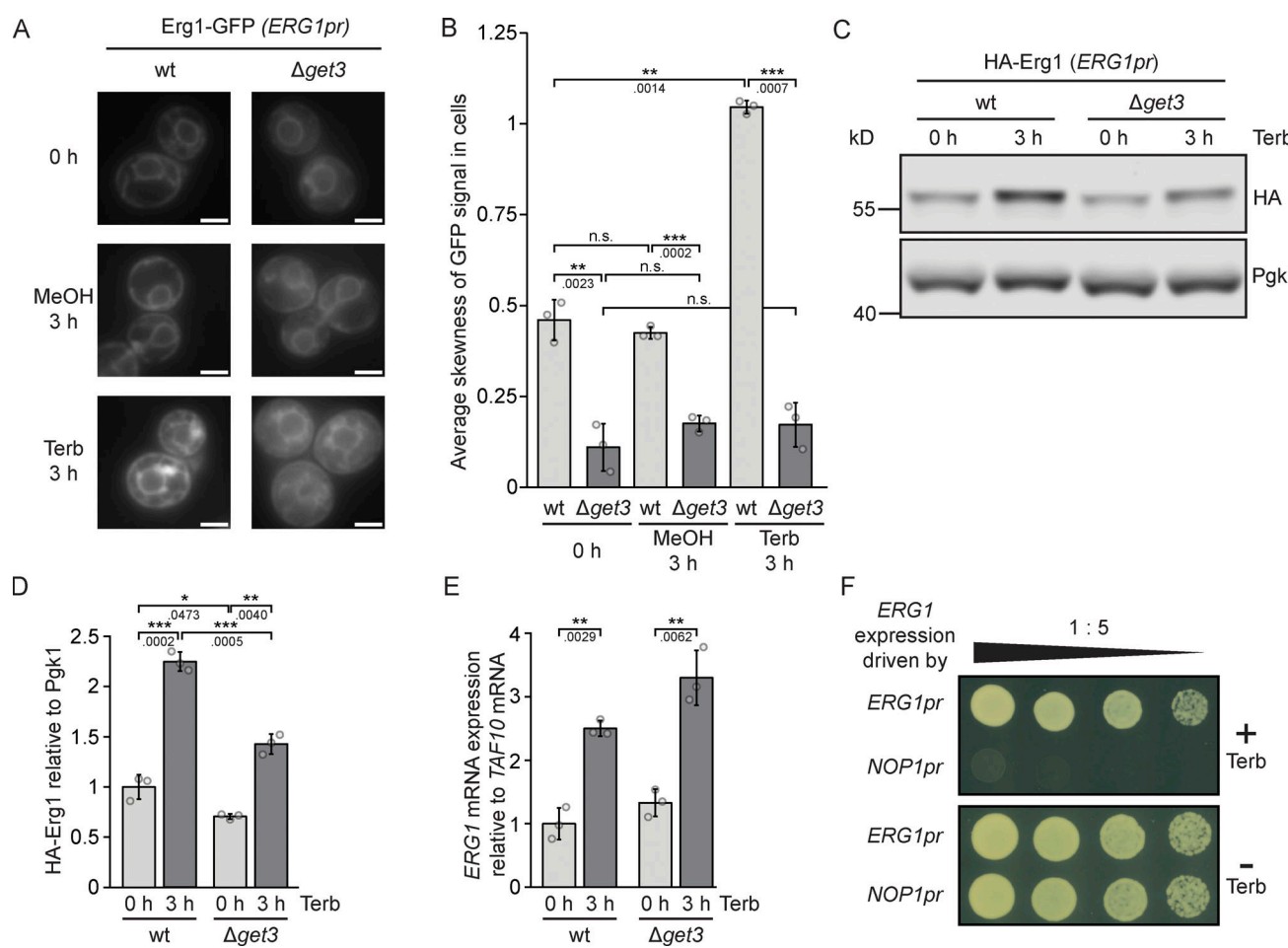

Figure 4. **Lack of Get3 impairs the induction of Erg1 protein following a terbinafine-induced pulse of *ERG1* mRNA expression. (A)** Fluorescence microscopy images of WT and Δ*get3* strains expressing genomically C-terminally GFP-tagged Erg1. Cells were imaged at the start of the experiment and 3 h after the addition of either 50 µg/ml terbinafine or as a control, methanol. Images are representative of three biological replicates with >100 cells imaged for each replicate. Scale bar: 2 µM. **(B)** Quantification of A. Bars represent the average of three biological replicates with individual data points shown as gray dots. Each point represents the average skewness of the distribution of the GFP signal in 100 cells from each sample as shown in A. Error bars indicate the standard deviation of the mean. The P values calculated using the two-sided Welch's *t* test are shown with numbers and represented as follows: ** < 0.01; *** < 0.001. **(C)** Immunoblot of cell lysates from WT and Δ*get3* strains expressing genomically N-terminally HA-tagged Erg1 before and after addition of terbinafine. Pgk1 serves as a loading control. Images are representative of three biological replicates. **(D)** Quantification of C. Bars represent the average of three biological replicates and individual data points are shown as gray dots. Error bars indicate standard deviation of the mean. The P values calculated using the two-sided Welch's *t* test are shown with numbers and represented as follows: ** < 0.01; *** < 0.001. **(E)** RT-qPCR measurement of the expression level of *ERG1* relative to the housekeeping gene *TAF10* in WT and Δ*get3* cells. Total RNA extracted from WT and Δ*get3* strains before and after the addition of terbinafine was converted to cDNA and used as a template for the PCR reaction. Expression was normalized according to the housekeeping mRNA *TAF10*. Bars represent the average of three biological replicates with individual data points shown as gray dots, each of which is the average of three technical replicates. Error bars indicate standard deviation of the mean. The P values calculated using the two-sided Welch's *t* test are shown with numbers and represented as follows: ** < 0.01. **(F)** Plate growth assay of strains in which the expression of *ERG1* is under control of either its endogenous promoter (*ERG1pr*) or the promoter of *NOP1* (*NOP1pr*). Strains were spotted out in a one-to-five dilution series on YPD plates with (+ Terb) and without (– Terb) 50 µg/ml terbinafine. The images are representative of three biological replicates. Source data are available for this figure: SourceData F4.

Erg1 protein levels, we sought to address the interplay between Get3 and Doa10 by analyzing the subcellular localization of N-terminally GFP-tagged Erg1 in strains lacking *DOA10*, *GET3*, or both genes (Fig. 5 B). Compared to the WT, in Δ*get3* cells, a diffuse GFP–Erg1 staining pattern was observed as in the initial screen (Fig. 1 D), whereas in the Δ*doa10* strain, GFP–Erg1 presented a stronger signal encompassing ER and LDs, reflecting the stabilization of the Erg1 protein. Co-deletion of *DOA10* and *GET3*, however, resulted in the stabilization of an ER-resident population of GFP–Erg1 compared to Δ*get3* cells, but the signal

appeared markedly weaker than in the Δ*doa10* strain. To test whether these changes are also reflected in the protein levels, we analyzed the amount of N-terminally HA-tagged Erg1 expressed from its genomic locus by Western blotting (Fig. 5, C and D). In both WT and Δ*get3* backgrounds, deletion of *DOA10* increased the steady-state levels of HA-Erg1, consistent with the fact that the enzyme is a substrate of the E3 ligase. However, *GET3* deletion reduced the amount of HA–Erg1 present to ~75%, irrespective of the presence or absence of *DOA10*. Consistent with previous results showing that drugs targeting specific steps

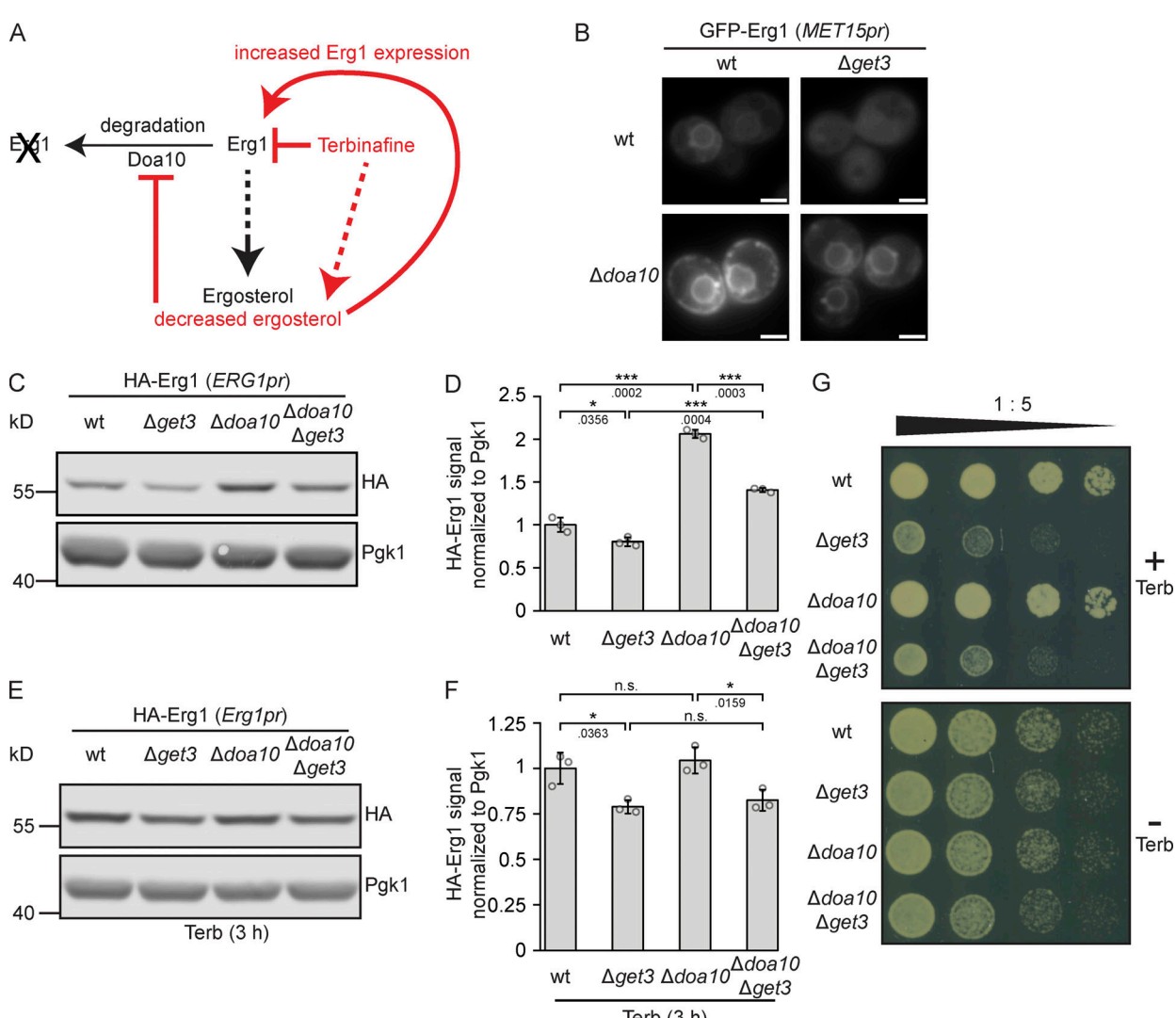

Figure 5. **Doa10 and Get3 affect Erg1 levels independently. (A)** Schematic representation of the effects of terbinafine and Doa10 on the stability and expression level of Erg1. **(B)** Fluorescence microscopy images of WT, Δget3, Δdoa10, and Δdoa10Δget3 strains expressing N- or C-terminally GFP-tagged Erg1 ectopically from the *MET15* promoter (*MET15pr*). Images are representative of three biological replicates with >200 cells imaged for each replicate. Scale bar: 2 μM. **(C)** Immunoblot of cell lysates from WT, Δget3, Δdoa10, and Δdoa10Δget3 strains expressing genomically N-terminally HA-tagged Erg1. Pgk1 served as a loading control. Images are representative of three biological replicates. **(D)** Quantification of C. Bars represent the average of three biological replicates with individual data points shown as gray dots. Error bars indicate standard deviation of the mean. The P values calculated using the two-sided Welch's *t* test are shown with numbers and represented as follows: * < 0.05; *** < 0.001. **(E)** Immunoblot of cell lysates from WT, Δget3, Δdoa10, and Δdoa10Δget3 strains expressing genomically N-terminally HA-tagged Erg1 after treatment with terbinafine (Terb). Pgk1 served as a loading control. Images are representative of three biological replicates. **(F)** Quantification of (E). Bars represent the average of three biological replicates with individual data points shown as gray dots. Error bars indicate standard deviation of the mean. The P values calculated using the two-sided Welch's *t* test are shown with numbers and represented as follows: * < 0.05. **(G)** Plate growth assay of WT, Δget3, Δdoa10 and Δdoa10Δget3 strains spotted out in a one-to-five dilution series on YPD plates with (+) or without (−) terbinafine (Terb). The images are representative of three biological replicates. Source data are available for this figure: SourceData F5.

of isoprenoid synthesis largely abolish the Doa10-dependent degradation of Erg1 (Foresti et al., 2013), treatment with terbinafine negated the effect of Δdoa10 on Erg1 protein levels in both the WT and Δget3 strains (Fig. 5, E and F). Therefore, as the loss of *DOA10* only increased Erg1 levels under normal growth conditions, but not when cells were treated with terbinafine, loss of *DOA10* should not be able to alleviate the terbinafine sensitivity of Δget3 cells. To test this, the growth of strains lacking *GET3*, *DOA10*, or both on a medium containing terbinafine was compared and, as expected, no differences in the

growth of the Δget3 and Δget3Δdoa10 strains were observed (Fig. 5 G). We conclude that Get3 and Doa10 play subsequent and hence independent roles in the biogenesis or degradation of Erg1, respectively. Both processes contribute to Erg1 levels under normal conditions. However, at low sterol levels, such as those elicited by treatment with terbinafine, Get3 action is required to ensure efficient targeting of the surge of newly synthesized squalene monooxygenase, whereas Erg1 becomes inaccessible to Doa10 under these conditions (Foresti et al., 2013).

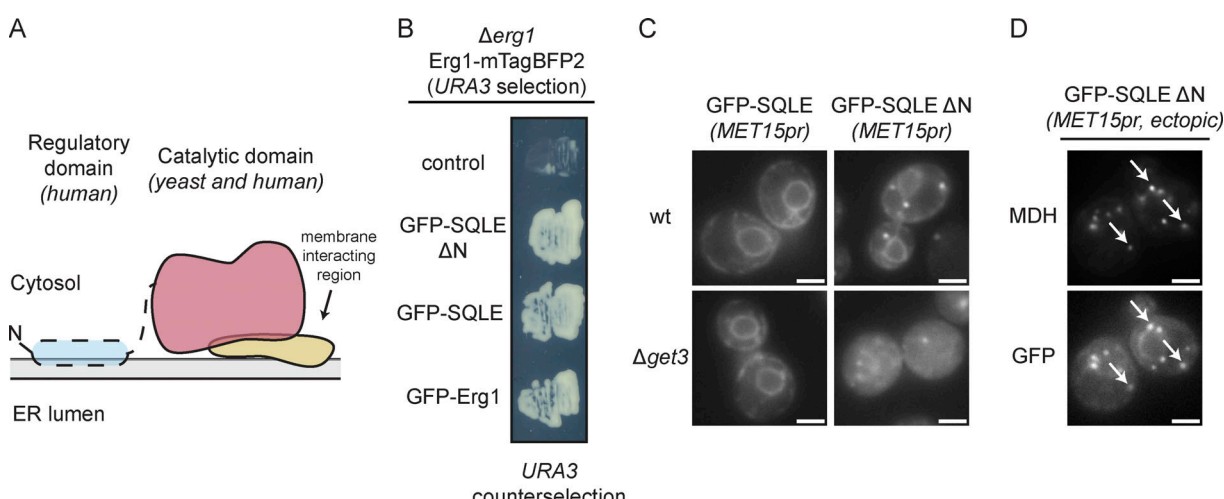

**Figure 6. Human SQLE can complement Erg1 and its processed form requires Get3 for proper ER and lipid droplet localization. (A)** Schematic representation of the structure of human SQLE. **(B)** Full-length (SQLE) and processed (SQLE ΔN) human SQLE can complement lack of *ERG1*. An Δ*erg1* strain carrying a URA3 marker-containing plasmid for expression of Erg1-mTagBFP2 was transformed with plasmids for the expression of N-terminally GFP-tagged full length human SQLE, processed SQLE (SQLE ΔN) or yeast Erg1 and individual colonies were streaked out onto a synthetic dropout plate containing 5-FOA to drive counterselection of the Erg1-mTagBFP2 plasmid. The image is representative of five colonies streaked out for each plasmid. **(C)** Fluorescence microscopy images of WT and Δ*get3* strains expressing N-terminally GFP-tagged full length (SQLE) or N-terminally truncated (SQLE ΔN) SQLE ectopically under control of the *MET15* promoter (*MET15pr*). Images are representative of three biological replicates with >200 cells imaged for each replicate. Scale bar: 2 μM. **(D)** Cells expressing N-terminally GFP-tagged SQLE ΔN were stained with the lipid droplet marker MDH. Images are representative of three biological replicates with >100 cells imaged for each replicate. Arrows indicate co-localization. Scale bar: 2 μM.

## The sterol sensor element of human SQLE renders the enzyme Get3-independent

Squalene monooxygenase is a conserved protein, but mammalian SQLE (Fig. 6 A) contains an ~100-amino-acid N-terminal domain not present in yeast Erg1, which contacts the ER membrane and functions as a sterol sensor (Gill et al., 2011). SQLE can be proteolytically processed in human cells, resulting in the loss of the N-terminal extension and giving rise to a full-length and a processed form in vivo (Coates et al., 2021). The processed version resembles yeast Erg1 and was found to be constitutively active. To investigate whether the high degree of conservation in the sterol synthesis pathway (Kachroo et al., 2015) extends to a role of Get3 in targeting human SQLE, we employed the 5-FOA based *URA3* counterselection method to test whether full-length and truncated SQLE can functionally complement yeast Erg1. The Δ*erg1* strain used in this assay was transformed with a second plasmid driving the expression of GFP-tagged full-length human SQLE or a shorter version lacking the N-terminal sterol sensing region, thus corresponding to the truncated form found in human cells in vivo. Counterselection on 5-FOA containing medium revealed that the expression of either form of the human protein supported the growth of Δ*erg1* cells comparably to Erg1 (Fig. 6 B). This result confirms previous reports that full-length SQLE can replace Erg1 (Jandrositz et al., 1991; Kachroo et al., 2015; Satoh et al., 1993) and demonstrates that the truncated form of SQLE is functional in budding yeast as well. Next, we addressed the Get3-dependence of the subcellular localization of full-length and truncated SQLE (Fig. 6 C). While full-length SQLE localized to the ER irrespective of the presence of Get3, the N-terminally truncated, Erg1-like form was strongly Get3-dependent in its ER localization. In yeast, only SQLE

lacking the N-terminal regulatory domain was observed in LDs (Fig. 6 D), consistent with a monotopic topology mediated by the immersion of the C-terminal helices (Fig. 3, A and B) into the LD lipid monolayer. Thus, we conclude that the Get3-dependence of the C-terminal domain of squalene monooxygenase is conserved, but that the additional membrane anchor provided by the N-terminal domain is sufficient to abrogate Get3-dependent targeting.

## Get3 is required for the ER localization of several hairpin proteins in yeast

Our results show that the tandem hydrophobic helices of Erg1 mediate its interaction with Get3 and that C-terminally GFP-tagged Erg1 requires Get3 for proper biogenetic targeting. Together with the fact that ER targeting of the truncated form of human SQLE was Get3 dependent, this raised the possibility that other proteins with a similar hairpin topology could also be clients of Get3. To this end, proteins that have two hydrophobic, putative TMSs less than ten amino acids apart anywhere in their sequence were identified based on the consensus of several different protein topology prediction algorithms (Weill et al., 2019). This yielded a list of 22 hairpin proteins that represent potential novel GET pathway clients (Table S3). Tsc10, which has been experimentally validated to contain a hairpin at its C-terminus (Gupta et al., 2009), and Ubx2, a monotopic membrane protein based on its localization to LDs (Wang and Lee, 2012), which may also contain a hairpin according to one of the tested prediction algorithms, were also considered. Strains from the SWAT library expressing *NOP1*-promoter-driven N-terminally GFP-tagged versions of these proteins (Weill et al., 2018) in the WT and Δ*get3* backgrounds were

imaged. From the 24 proteins tested (Fig. S5 and Table S3), five showed clearly different localization patterns in the absence of Get3 (Fig. 7 A). Lam1 and Sip3 are known to have low expression levels (Gatta et al., 2015; Weill et al., 2018), reflected in a weak ER signal, just visible above the background fluorescence of the cells. However, in Δget3, this ER staining was lost. Similarly, in cells lacking Get3, the ER signal of Prm9 became largely diffuse. Tsc10 and Ubx2 also partially relocalized from the ER and appeared in cytosolic structures identified as mitochondria by co-staining cells with Mito-Tracker (Fig. 7 B). Although Ubx2 localizes to mitochondria to some extent in WT cells (Wang and Lee, 2012), the proportion of mitochondrial localized Ubx2 was markedly increased in Δget3 cells. As both mislocalization to the cytosol and mislocalization to mitochondria are hallmarks of Get3 clients in the absence of a functional GET-pathway (Li et al., 2019; Schuldiner et al., 2008), these data support the model that these proteins may be Get3 clients. Strikingly, in contrast to Erg1, the hairpins in these proteins are not located at their C-termini, except for Tsc10 (Fig. 7 C). This is in line with our finding that it is not the C-terminal position of the Erg1 hairpin that makes it a Get3 client (Fig. 3 K). To assess whether the mislocalization of these proteins results in altered protein levels in Δget3 cells, the steady-state levels of the potential Get3 clients were determined by immunoblotting (Fig. 7, D and E). Although Tsc10 and Ubx2, which mislocalize to mitochondria in Δget3 cells and may be stabilized there, showed no quantitative changes, Lam1 and Sip3 both displayed significantly decreased levels in the absence of Get3, likely reflecting the degradation of the non-ER integrated population. To corroborate that the observed mislocalization occurs during biogenesis and to further consolidate these proteins as Get3 targeting clients, the identified proteins were expressed from the inducible GAL1 promoter in WT and Δget3 strains. As expected, the mislocalization of these proteins was exacerbated by their strong, transient overexpression (Fig. 7 F). Interestingly, aggregate-like foci containing Lam1, Sip3, or Prm9 were observed in the Δget3 strain. Mislocalized Get3 clients have previously been observed to accumulate in chaperone-rich protein aggregates containing the cytosolic chaperone Hsp104 (Powis et al., 2013). Therefore, Hsp104 was fluorescently tagged, and we confirmed that Lam1, Sip3, and Prm9 all co-localize with Hsp104-marked aggregates when transiently overexpressed (Fig. 7 G). In conclusion, the proteins identified in our screen likely represent further Get3 clients containing a hairpin, instead of a TA, and delineate a novel class of GET pathway clients.

## Discussion

Many phenotypes of get mutants relate to cellular stress conditions, such as oxidative or heat stress, or toxic fatty acids (Metz et al., 2006; Ruggles et al., 2014; Schuldiner et al., 2008; Shen et al., 2003). However, identifying specific GET pathway clients as the underlying cause of the observed phenotypes has been challenging. Our results not only expand the client spectrum of the GET pathway to include the hairpin protein Erg1 but

also provide a plausible explanation for sterol-related phenotypes of Δget strains, such as liposensitivity and reduced LD content (Ruggles et al., 2014). Knowledge of the biogenesis of different classes of membrane proteins has expanded considerably in recent years (Aviram and Schuldiner, 2017; Dudek et al., 2015; O'Keefe et al., 2021), but little is still known about the biogenetic processes that enable monotopic membrane proteins to integrate into phospholipid mono- and bilayers (Allen et al., 2019; Dhiman et al., 2020). The discovery of Erg1 as a biogenetic client of Get3 highlights the GET pathway as a potential targeting route for proteins of this class. Indeed, the ability of the GET pathway components to chaperone the hydrophobic TMS of TA proteins and integrate them into the ER membrane is well suited to protect the hydrophobic patches of proteins containing hairpins similar in size to TMSs and to facilitate their passage into the hydrophobic core of the ER membrane (Allen et al., 2019; Blobel, 1980).

The localization of Erg1 to LDs and the available structure of its human homolog indicate that Erg1 can be a monotopic integral membrane protein and our results also indicate that it is not a TA protein. However, it is possible that the two C-terminal helices may also assume a transmembrane conformation at the ER placing the C-terminus in the cytosol as their length would be sufficient. Intriguingly, our finding that several other proteins that contain hairpins of a similar size mislocalize in cells lacking Get3 suggests that Get3 can act as a targeting factor for proteins associated with membranes via a single hairpin composed of two longer hydrophobic helices. Two of these, Lam1 and Sip3, are also involved in sterol transfer between membranes (Gatta et al., 2015), while Tsc10 is an enzyme of sphingolipid metabolism (Beeler et al., 1998). Thus, their mislocalization in the absence of Get3 could further contribute to the liposensitivity and perturbed lipid homeostasis in Δget3 cells. Notably, the hydrophobic helices of Tsc10 are thought to assume a transmembrane conformation as well as a monotopic one (Gupta et al., 2009). As this may also be the case for Erg1, it is conceivable that the other proteins identified here as potential Get3 clients could more generally assume dual topologies and that Get3 has a broader role as a targeting factor for this type of protein. The fact that other hairpin proteins that mislocalized in Δget3 were not specifically identified in our mass spectrometry analysis likely reflects the low expression levels of some of these proteins (Weill et al., 2018) and the challenges of comprehensive recovery of binding partners of proteins with broad interactomes like Get3. Furthermore, the stability of the Get3-client complex may be different for each client, which, along with the saturation of Get3 with preferred clients and other targeting pathways, taking care of a portion of the flux of de novo synthesized client proteins, may limit the number of recoverable targeting clients in co-immunoprecipitation experiments.

The hydrophobic helices of the hairpin of these proteins need to be recognized by the pretargeting complex composed of Get4, Get5, and Sgt2 and transferred to Get3. Based on the finding that the introduction of hydrophilic residues within the hydrophobic groove of Get3 disrupts interactions with Erg1, it is possible that the hairpin could be similarly accommodated within the Get3 protein as the TMS of TA protein clients. Indeed, our results

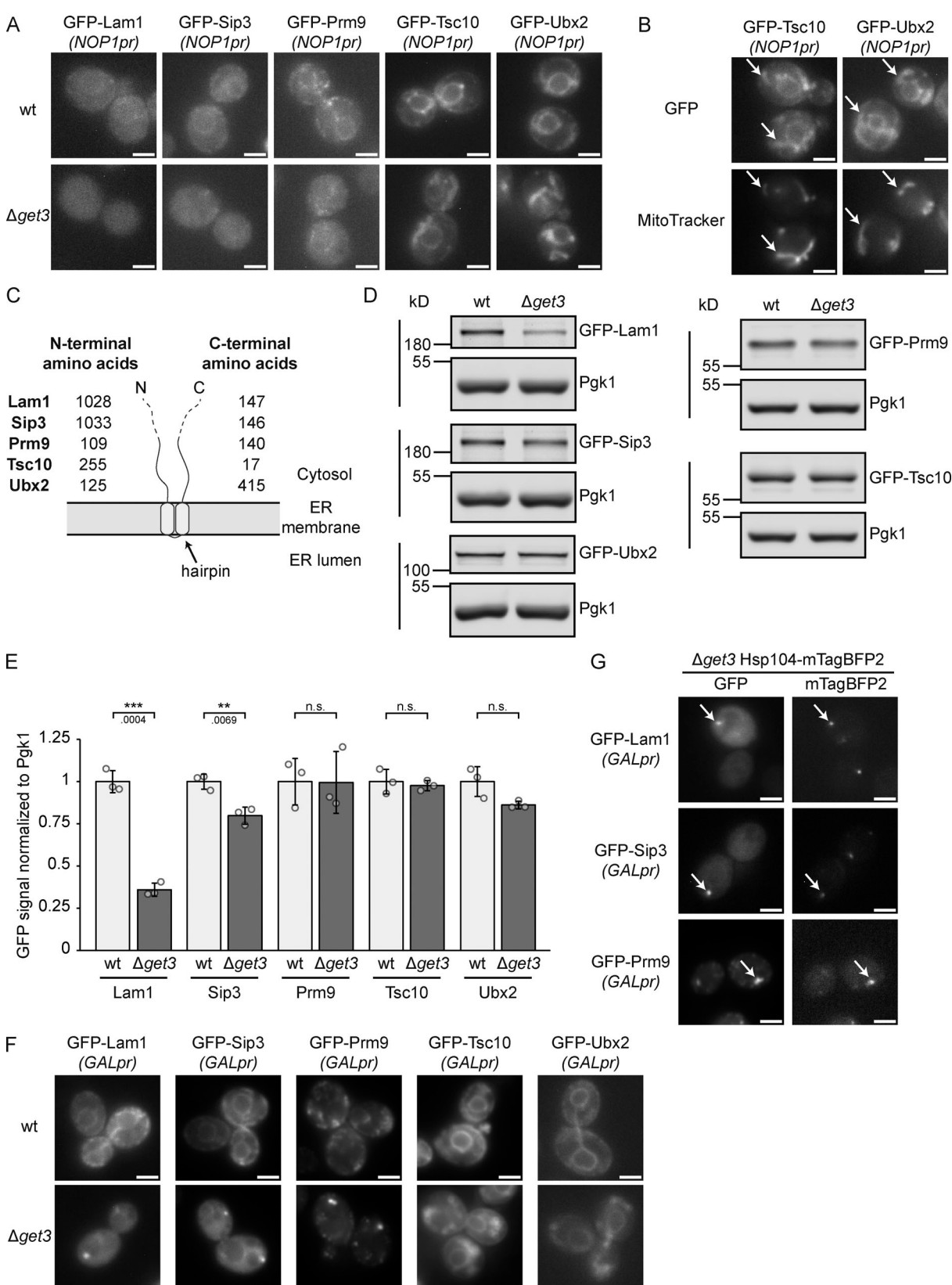

Figure 7. **Several hairpin proteins behave as Get3 clients in yeast. (A)** Fluorescence microscopy images of WT and Δget3 strains expressing the indicated genomically N-terminally GFP-tagged proteins under the control of the *NOP1* promoter (*NOP1pr*). Images are representative of three biological replicates with >100 cells imaged for each replicate. Scale bar: 2 µM. **(B)** Fluorescence microscopy images of Δget3 strains expressing the indicated genomically N-terminally GFP-tagged proteins under the control of the *NOP1* promoter (*NOP1pr*). Cells were stained with MitoTracker Orange (MitoTracker) to visualize mitochondria. Images are representative of three biological replicates with >100 cells imaged for each replicate. Arrows indicate co-localization. Scale bar: 2 µM.

**(C)** Schematic representation of the topology of proteins visualized in A. The number of amino acid residues present N- or C-terminally of the predicted hairpins are indicated for each protein. **(D)** Immunoblot of cell lysates from WT and Δget3 strains expressing the indicated genomically N-terminally GFP-tagged proteins under the control of the *NOP1* promoter (*NOP1pr*). Pgk1 served as a loading control. Images are representative of three biological replicates. **(E)** Quantification of D. Bars represent the average of three biological replicates with individual data points shown as gray dots. Error bars indicate standard deviation of the mean. The P values calculated using the two-sided Welch's *t* test are shown with numbers and represented as follows: ** < 0.01; *** < 0.001. **(F)** Fluorescence microscopy images of WT and Δget3 strains expressing *GAL1* promoter-driven N-terminally GFP-tagged Lam1, Sip3, Prm9, Tsc10 ectopically and Ubx2 genomically. Strains were grown in synthetic dropout media containing 2% raffinose and imaged 1 h after transition to media containing 2% galactose (Gal). Images are representative of three biological replicates with >200 cells imaged for each replicate. Scale bar: 2 μM. **(G)** Fluorescence microscopy images of Δget3 strains expressing genomically C-terminally mTagBFP2-tagged Hsp104 and *GAL1* promoter-driven N-terminally GFP-tagged Lam1, Sip3, Prm9 ectopically. Strains were grown in synthetic dropout media containing 2% raffinose and imaged 1 h after transition to media containing 2% galactose (Gal). Images are representative of three biological replicates with >200 cells imaged for each replicate. Arrows indicate co-localization. Scale bar: 2 μM. Source data are available for this figure: SourceData F7.

show that Get3 recognizes both helices of the Erg1 hairpin. However, the effect of the deletion of individual helices is not additive, as the loss of the more hydrophobic helix almost completely abolishes the interaction between Erg1 and Get3, whereas loss of the other one does not. This suggests that the hairpin is recognized at least to some extent differently from single hydrophobic helices. Since Get3 and its evolutionary homologs are known to be able to form multimeric complexes capable of client binding (Bozkurt et al., 2009; Suloway et al., 2012), it is possible that a multimeric form of Get3 is involved in the binding of helices arranged as a hairpin. To insert into the membrane, the hairpin also needs to bypass the charged surface of the membrane and reach its hydrophobic core. Thus, similarly to TA protein targeting, the Get1/2 receptor complex may be able to provide a conduit for the hydrophobic helices of hairpin proteins once delivered by Get3.

It has emerged that despite being one of the major biogenetic pathways for TA proteins, only a few TA proteins show defective steady-state localization in the absence of a functional GET pathway (Li et al., 2019; Rivera-Monroy et al., 2016). It is, therefore, possible that more of the proteins predicted to contain a single hairpin composed of long hydrophobic helices (Table S3) actually use the GET pathway, but the effect of loss of Get3 is masked by the presence of alternative targeting pathways (Aviram et al., 2016). Indeed, the redundancy between different targeting pathways has emerged as an important factor for explaining why, under normal growth conditions, the loss of a specific targeting pathway, such as the GET pathway, is well tolerated by the cells (Aviram et al., 2016; O'Keefe et al., 2021). However, our results suggest that high-fidelity and high-efficiency membrane protein targeting, guaranteed by dedicated pathways, becomes necessary when cells are challenged by the requirement for increased biogenetic pulses of particular proteins. In the case of Erg1, the rate-limiting enzyme controlling the flow of isoprene units toward steroids is subject to sterol-dependent regulation by product-mediated feedback inhibition. Under normal growth conditions, sufficient Erg1 reaches the ER membrane to sustain the growth of Δget3 cells. However, sterol depletion via terbinafine treatment upregulates the expression of *ERG1* mRNA and results in a biogenetic pulse of Erg1, which renders Get3 essential to ensure efficient delivery of the newly synthesized protein to the ER membrane. This finding emphasizes the fact that approaches addressing the specific client spectra of different ER targeting pathways should

conceptually include expression kinetics as a highly relevant parameter. Both adaptations to cellular stress and development in multicellular organisms entail gene expression programs that rapidly upregulate the production of specific proteins at particular times. We propose that the GET pathway becomes essential in yeast and multicellular organisms under conditions where its clients require particularly efficient biogenesis. This model reconciles the viability of tissues or cell lines lacking GET/TRC40 pathway components (Casson et al., 2017) with the early embryonic lethality caused by loss of TRC40 (Mukhopadhyay et al., 2006; Tran et al., 2003; also compare Table 2 in Borgese et al. [2019]).

Our findings further suggest that alongside the high functional homology between yeast Erg1 and human SQLE, evolutionary conservation could extend to the biogenetic dependency on the GET pathway. The LD localization of the processed form of SQLE in yeast (Fig. 6 C) indicates that SQLE, similar to Erg1, can be a monotopic membrane protein. This is also in line with the topology present in the available structural model (Padyana et al., 2019). Although the ER and LD localization of the processed form of SQLE was strongly Get3-dependent in yeast, full-length SQLE localized exclusively to the ER and was not affected by lack of Get3. This suggests that the N-terminal cholesterol sensing element of SQLE further anchors the protein to the membrane and makes full-length SQLE Get3-independent in yeast cells. However, it is possible that the different lipid environment of human cells and/or proteolytic processing of full-length SQLE to remove the N-terminal membrane anchor necessitate ER targeting by TRC40 in certain conditions.

The newly discovered function of the GET pathway in targeting Erg1 and potentially other hairpin proteins to the ER, which becomes especially important in conditions when a rapid pulse of protein expression is required, provides evidence supporting a mechanistic role for the GET pathway in the dynamic regulation of sterol metabolism. Building on this finding, it will be interesting to see whether the importance of GET pathway targeting is similarly increased when the biogenetic requirements of other client proteins are altered. Furthermore, the recognition of a broader client spectrum of the GET pathway to include hairpin proteins opens the door to the identification of other non-canonical clients, thereby providing a more comprehensive understanding of the role of the GET/TRC40 pathways in protein homeostasis in single and multicellular organisms.

## Materials and methods

### Plasmids and cloning methods

All plasmids used in this study are listed in Table S4. pRS415 *MET15pr*, pRS416 *MET15pr*, and p426 *GAL1pr* were used as backbones for plasmids used in this study (Mumberg et al., 1994). Plasmids were generated using standard cloning methods, including Gibson assembly (Gibson et al., 2009) or T4 ligation, as indicated for each plasmid. For T4 ligation, inserts were amplified by PCR, then digested for 1 h at 37°C along with the plasmid backbone with appropriate restriction enzymes (Thermo Fisher Scientific). After agarose gel electrophoresis, DNA fragments were purified using High Pure PCR Product Purification Kit (Roche). The plasmid backbone and inserts were ligated with T4 ligase (Thermo Fisher Scientific). *Escherichia coli* ElectroTenBlue (Agilent) cells were electroporated with the ligation mixture, plated onto appropriate selection plates, and incubated for 1 d at 37°C. Individual colonies from the plate were used to amplify plasmids, which were confirmed by Sanger sequencing. For Gibson assembly, the plasmid backbone was prepared as described for T4 ligation and then fused with PCR-amplified inserts in Gibson assembly reaction mixture for 1 h at 50°C. Starting with the electroporation of bacterial cells, cloning proceeded identically to the T4 ligation method described above.

### Yeast strain generation

Unless indicated otherwise, all yeast strains used in this study are from the By4741 genetic background (Brachmann et al., 1998) and are listed along with all DNA oligomers used in their creation in Table S4. Antibiotic resistance markers were amplified from a plasmid derived from pCEV-G1 Km (Vickers et al., 2013) carrying the clonNAT and phleomycin resistance cassettes as well. The clonNAT resistance cassette used for tagging Hsp104 genomically with mTagBFP2 was derived from pAG25 (Goldstein and McCusker, 1999). Transformation with plasmids and genomic modification was done using the lithium acetate–polyethylene glycol method described previously (Gietz et al., 1992). Yeast cells expressing plasmids were transformed freshly before each experiment in the appropriate genomic background. For plasmid transformation and genomic modifications with auxotrophic selection markers, yeast strains were incubated for 2 d at 30°C on synthetic dropout plates lacking appropriate auxotrophic selection markers. Genomically modified strains with antibiotic selection markers were first allowed to grow on YPD plates for 1 d at 30°C before replica-copying them to YPD plates containing 200 μg/ml G418 (Sigma-Aldrich), 100 μg/ml clonNAT (HKI Jena), or 37.5 μg/ml Phleomycin (InvivoGen), according to the selection marker used. Genomically modified strains were then streaked out onto appropriate selection plates, and single colonies were verified using PCR or Western blotting.

To generate the Δ*erg1* pRS416 *MET15pr*::Erg1-mTagBFP2 strain, the By4741 strain carrying the plasmid was used to replace *ERG1* with the *NAT* resistance cassette due to the *ERG1* gene being essential.

HA–*ERG1* strain was created by transforming Δ*erg1* pRS416 *MET15pr*::Erg1-mTagBFP2 with a cassette encoding HA–*ERG1* containing ~40 base pairs on both 5′ and 3′ ends homologous to the genomic sequence directly adjacent to the *ERG1* gene. After growth on a YPD plate, the yeast was replica-copied onto a YPD plate containing 1 mg/ml 5-fluoroorotic acid (Thermo Fisher Scientific) to select colonies capable of losing the pRS416 *MET15pr*::Erg1-mTagBFP2 plasmid indicating correct genomic integration of the HA–*ERG1* cassette.

Yeast strains for the expression of N-terminally GFP-tagged proteins under the control of the *NOP1* promoter were selected from the previously published SWAT library (Weill et al., 2018). Strains lacking *GET3* were generated with a synthetic genetic array (SGA)-based method using an SGA-compatible *MATα* strain as described previously (Tong and Boone, 2007). Briefly, SWAT library strains were mated with the SGA-compatible *MATα* strain lacking *GET3*. The resulting diploid cells were selected and induced to sporulate. Progeny carrying all marker genes and lacking *GET3* were selected on synthetic dropout plates containing clonNAT and lacking appropriate auxotrophic markers.

### Yeast growth conditions

All experiments were conducted with yeast growing logarithmically. For microscopy and in all cases when the strain carried a plasmid, appropriate synthetic dropout media was used. All other experiments used yeast grown in YPD media unless indicated otherwise. To test the effect of terbinafine on the expression of Erg1-GFP from its endogenous locus, YPD media was used. Terbinafine (Sigma-Aldrich) dissolved in methanol was applied at a final concentration of 50 μg/ml.

### Plate growth assays

Logarithmically growing strains were normalized to $OD_{600}$ 0.8. A total of 5 μl of the normalized culture was spotted out in a 1:5 dilution series onto YPD plates with the appropriate composition. Strains containing plasmids were spotted onto synthetic dropout plates lacking the appropriate auxotrophic marker. Control plates for growth on terbinafine (Sigma-Aldrich) contained an equivalent amount of methanol instead. Plates were incubated at 30°C for 1 d, except for terbinafine-containing plates, which were incubated for 2 d to account for generally slower growth of yeast in presence of the inhibitor.

### *URA3* counterselection assay

After the transformation of the test strain with the indicated pRS415-derived plasmids, single colonies were streaked out onto a synthetic dropout plate lacking leucine containing 1 mg/ml 5-fluoroorotic acid (Thermo Fisher Scientific) and allowed to grow for 3 d at 30°C.

### Galactose-induced protein expression

Yeast was grown to logarithmic phase in synthetic dropout media lacking uracil containing 2% raffinose instead of D-glucose. Synthetic dropout media lacking uracil containing 10% galactose was added to obtain a final concentration of 2% galactose in the culture, and the culture was allowed to grow for 1 h before imaging the strains.

### Microscopy

For the screens of N- and C-terminally GFP-tagged proteins, stationary-phase cells were diluted into 384-well glass-bottom

microtiter plates (Brooks Life Science Systems) containing low fluorescence minimal media (FORMEDIUM Ltd.) supplemented with 2% glucose, methionine, histidine, uracil, and leucine, and allowed to grow for 4 h at 30°C. For all other images, strains were grown to logarithmic phase in an appropriate selection media while shaking in tubes at 30°C. About 10 µl of the culture was diluted in 50 µl synthetic dropout media in 384-well glass-bottom microtiter plates (Brooks Life Science Systems) and imaged immediately.

For lipid droplet staining, MDH (abcepta) was added to cultures to a final concentration of 2 µM from a 200 µM stock in dimethylsulfoxide, incubated for 5 min, and imaged immediately.

For mitochondrial staining, MitoTracker Orange CMTMRos (Molecular Probes) was added to cultures to a final concentration of 100 nM from a 10 µM stock in dimethylsulfoxide, incubated for 20 min and imaged immediately.

Images were acquired at room temperature with a Nikon Ti2 2-E inverted microscope equipped with a computer-controlled stage, a Lumencor Spectra X light source, a pco.edge 5.5 M-AIR-CL-PCO sCMOS camera (2,560 × 2,156 pixels) using NIS-Elements software (Nikon). The focal plane was detected with the Perfect Focus System (Nikon). Images were acquired with a CFI Plan Apo Lambda 100×/1.45 oil objective using settings appropriate for GFP (Ex: 470, Em: 520/35), MitoTracker (Ex: 555, Em: MultiBand Filter 433/30 517/30 613/30), or MDH and mTagBFP2 (Ex: 395, Em: 433/24). Image cropping, brightness adjustment, and conversion from 16-bit to RGB encoding was done in ImageJ (Schneider et al., 2012) to generate figures.

To quantify the distribution of Erg1-GFP signal, 100 cells with a visible ER signal were selected for each sample. and the distribution of the pixel intensities was quantified using ImageJ (Schneider et al., 2012). The average of these cells was considered as an individual data point for each sample and used for comparison between samples.

### Protein extraction from microsomes
Microsomes were prepared and proteins extracted based on previously published protocols (Gable et al., 2000). Yeast cells growing logarithmically in YPD were harvested, washed with water and resuspended in 3 ml/g storage buffer (50 mM Tris, 1 mM EGTA, 1 tablet/20 ml cOmplete EDTA-free protease inhibitor cocktail [Roche], 1 mM DTT). Cells were lysed with glass beads by vortexing three times for 3 min with a 1-min incubation on ice between rounds. The lysate was transferred to new tubes and cell debris was pelleted by centrifugation at 500 $g$ for 10 min at 4°C. The resulting supernatant was centrifuged again at 500 $g$ for 5 min at 4°C. The supernatant was centrifuged at 100,000 $g$ for 50 min at 4°C, and the pellet was resuspended in a storage buffer with 33% glycerol equivalent to 50% of the volume before ultracentrifugation and then stored at –80°C. Protein concentration of the microsomes was determined with Pierce BCA protein assay (Thermo Fisher Scientific).

Proteins were extracted by incubating microsomes corresponding to 500 µg protein for 1 h on ice in storage buffer, 0.5 M NaCl, 2.5 M Urea, 0.1 M $Na_2CO_3$, 1% IGEPAL (Sigma-Aldrich), or 1% Triton X-100 (Sigma-Aldrich), followed by centrifugation at 100,000 $g$ for 50 min at 4°C. Proteins in the supernatant and the pellet were precipitated by incubating the samples with 12.5% trichloroacetic acid for 10 min on ice, then centrifuged at 15,000 $g$ for 10 min at 4°C. The protein pellet was washed twice with acetone, dried at room temperature, and resuspended in NuPAGE LDS Sample Buffer (Thermo Fisher Scientific) to a final protein concentration of 5 µg/µl. Samples corresponding to 25 µg protein were analyzed by immunoblotting.

### GFP–TEV immunoprecipitation
Strains were grown to $OD_{600}$ 0.8 in synthetic dropout media lacking appropriate auxotrophic selection markers. A total of 200 ml of each strain was harvested, and the cell pellet was stored at –80°C. Pellets were thawed and resuspended in 3 ml lysis buffer (25 mM Tris-HCL pH 7.5, 50 mM KCl, 10 mM $MgCl_2$, 2 mM DTT, 5% glycerol, 1 tablet/20 ml cOmplete EDTA-free protease inhibitor cocktail [Roche]), then crushed in a mortar with pestle under liquid nitrogen. Thawed lysates were centrifuged at 500 $g$ for 5 min at 4°C and the supernatant was transferred to a new tube and centrifuged at 500 $g$ for 5 min at 4°C. Protein concentration of the lysate was determined with Pierce BCA protein assay (Thermo Fisher Scientific) and 3 mg of total protein was incubated with 50 µl µMACS Anti-GFP MicroBeads (Miltenyi Biotec) while shaking for 1 h at 4°C. Samples were loaded onto µ Columns (Miltenyi Biotec) in a µMACS Separator (Miltenyi Biotec) and washed twice with 300 µl TEV cleavage buffer (50 mM Tris-HCl pH 7.5, 150 mM NaCl, 2 mM DTT). About 25 µl TEV cleavage buffer containing 5 µg TEV protease (custom-made) was added onto the columns and incubated for 1 h at room temperature. Input and eluate samples were mixed with NuPAGE LDS Sample Buffer (Thermo Fisher Scientific), incubated at 70°C for 5 min, and stored at –20°C until analysis by Western blotting or mass spectrometry.

### Mass spectrometry of GFP-TEV immunoprecipitates
Eluates from immunoprecipitation of TEV-GFP tagged proteins were resuspended in NuPAGE LDS Sample Buffer (Thermo Fisher Scientific), incubated at 70°C for 5 min, and resolved on NuPAGE 4–12% Bis-Tris gels. A total of 23 gel slices were cut from each gel using a custom-made cutter. Proteins in each gel fraction were reduced by incubating with 10 mM dithiothreitol at 56°C for 50 min, alkylated by incubation with 55 mM iodoacetamide for 20 min at RT in the dark, and digested using an MS-grade trypsin (Sigma-Aldrich). Digested peptides were extracted using acetonitrile and 5% formic acid and dried in a vacuum concentrator. Dried peptide samples were redissolved in 2% (v/v) acetonitrile (ACN), 0.1% (v/v) formic acid (FA) loading buffer and injected into a nano-LC system operated by UltiMate 3000 RSLC (Thermo Fisher Scientific). The LC was equipped with a C18 PepMap100-trapping column (0.3 × 5 mm, 5 µm; Thermo Fisher Scientific) connected to an in-house packed C18 analytical column (75 µm × 300 mm; Reprosil-Pur 120C18 AQ, 1.9 µm, Dr. Maisch GmbH). The LC was equilibrated using 5% (v/v) buffer B (80% [v/v] ACN 0.1% [v/v] FA in water) and 95% (v/v) buffer A (0.1% [v/v] FA in water). The peptides were eluted using the following gradient: (i) 10–45% linear increase of buffer B over 43 min; (ii) wash-out at 90% buffer B for 6 min; (iii) re-equilibration at 5% buffer B for 6 min. Eluting peptides were

sprayed into Q Exactive HF (Thermo Fisher Scientific) mass spectrometer operated in a data-dependent acquisition mode. MS1 scans of 350–1,600 m/z range were collected at 60,000 resolution and an automatic gain control (AGC) target of 1e6 and a maximum injection time (MaxIT) of 50 ms. Thirty most intense precursor ions of charge 2–5 were subjected to fragmentation using normalized collision energy (NCE) of 30%. MS2 scans were acquired at a resolution of 15,000, 1e5 AGC target, and MaxIT of 54 ms. The dynamic exclusion was set to 25 s. Of the identified *S. cerevisiae* proteins, potential contaminants and decoy peptides were removed along with proteins not identified in all three replicates of Get3 DE-TEV-GFP. Intensity values were $\log_2$ transformed, and the missing values were imputed with values based on a normal distribution (width: 0.3; down shift: 1.8; mode: separately for each column) using Perseus (Tyanova et al., 2016). Two-tailed Welch's *t* test was used to calculate the statistical significance of the difference in the enrichment of each protein in the Get3 DE-TEV-GFP and Get3 DE FIDD-TEV-GFP samples. Further analysis focused on proteins enriched statistically significantly ($P < 0.05$) differentially and on average at least eightfold between the Get3 DE-TEV-GFP and Get3 DE FIDD-TEV-GFP samples.

### Yeast cell lysis for protein analysis
Sample preparation was adapted from the previously described NaOH lysis protocol (Kushnirov, 2000). Briefly, 750 µl of logarithmically growing cells were pelleted and then resuspended in 1 ml 250 mM NaOH. Samples were incubated on ice for 10 min, pelleted for 1 min by centrifugation at 16,000 *g*, and resuspended in NuPAGE LDS Sample Buffer (Thermo Fisher Scientific) corresponding in µl to 100 × $OD_{600}$ of the NaOH solution containing the samples. After incubation at 70°C for 5 min, the samples were centrifuged at 16,000 *g* for 30 s and stored at –20°C until later use. A total of 7 µl was used for Western blot analysis.

### Western blotting
Samples were resolved in Bis-Tris gels and transferred onto PVDF membranes. The membrane was blocked in Tris-buffered saline (TBS) containing 5% milk for 1 h followed by incubation with primary antibodies in TBS containing 0.1% Tween-20 (Roth) overnight at 4°C. After rinsing the membranes in TBS containing 0.1% Tween-20, incubation with secondary antibodies followed in TBS containing 0.1% Tween-20 and 0.01% sodium dodecyl sulfate. Membranes were scanned in a LI-COR Odyssey scanner and quantified in Image Studio Lite 5.2.5 (LI-COR).

### Antibodies used in Western blotting
Primary antibodies used include polyclonal guinea pig anti-Get3 diluted 3,000-fold (Metz et al., 2006), polyclonal rabbit anti-Sed5 diluted 5,000-fold (Schuldiner et al., 2008), polyclonal rabbit anti-Ysy6 diluted 3,000-fold (generated using the peptide MAVQTPRQRLANAKFC and validated in Fig. S4 C, from Dobberstein laboratory), polyclonal rabbit anti-HA diluted 5,000-fold (#ab9110; Abcam), and monoclonal mouse IgG1 anti-Pgk1 diluted 5,000-fold (#459250; Invitrogen).

Secondary antibodies were diluted 10,000-fold and included IRDye 680LT anti-mouse IgG1 (#926-68050; LI-COR), IRDye

680LT anti-rabbit (#926-68023; LI-COR), IRDye 800CW anti-rabbit (#926-32213; LI-COR), and IRDye 800CW anti-guinea pig (#926-32411; LI-COR).

### Yeast lipidome analysis
Fifteen $OD_{600}$ units of logarithmically growing cells in SC or YPD media were pelleted and washed three times in 155 mM ammonium bicarbonate buffer. Cell pellets were snap-frozen in liquid nitrogen and stored at –80°. Pellets thawed on ice were resuspended in 155 mM ammonium bicarbonate buffer and lysed with glass beads by vortexing three times for 5 min at 4°C, with short pauses between each round. The lysate was snap-frozen in liquid nitrogen and stored at –80°. After thawing, samples were pelleted at 350 g for 2 min, the supernatant was taken, frozen, and MS-based lipid analysis was performed by Lipotype GmbH as described (Ejsing et al., 2009; Klose et al., 2012). Briefly, lipids were extracted using a two-step chloroform/methanol procedure (Ejsing et al., 2009). Samples were spiked with internal lipid standard mixture containing: cytidine diacylglycerol 17:0/18:1, ceramide 18:1;2/17:0, diacylglycerol 17:0/17:0, lysophosphatidate 17:0, lyso-phosphatidylcholine 12:0, lysophosphatidylethanolamine 17:1, lyso-phosphatidylinositol 17:1, lysophosphatidylserine 17:1, phosphatidate 17:0/14:1, phosphatidylcholine 17:0/14:1, phosphatidylethanolamine 17:0/14:1, phosphatidylglycerol 17:0/14:1, phosphatidylinositol 17:0/14:1, phosphatidylserine 17:0/14:1, ergosterol ester 13:0, triacylglycerol 17:0/17:0/17:0, stigmastatrienol, inositolphosphorylceramide 44:0;2, mannosylinositolphosphorylceramide 44:0;2 and mannosyl-di-(inositolphosphoryl)ceramide 44:0;2. After extraction, the organic phase was transferred to an infusion plate and dried in a speed vacuum concentrator. First step dry extract was resuspended in 7.5 mM ammonium acetate in chloroform/methanol/propanol (1:2:4, V:V:V) and second step dry extract in 33% ethanol solution of methylamine in chloroform/methanol (0.003:5:1; V:V:V). All liquid handling steps were performed using Hamilton Robotics STARlet robotic platform with the Anti Droplet Control feature for organic solvents pipetting. Samples were analyzed by direct infusion on a QExactive mass spectrometer (Thermo Fisher Scientific) equipped with a TriVersa NanoMate ion source (Advion Biosciences). Samples were analyzed in both positive and negative ion modes with a resolution of Rm/z = 200 = 280,000 for MS and Rm/z = 200 = 17,500 for MSMS experiments, in a single acquisition. MSMS was triggered by an inclusion list encompassing corresponding MS mass ranges scanned in 1-D increments (Surma et al., 2015). Both MS and MSMS data were combined to monitor ergosterol ester, diacylglycerol, and triacylglycerol ions as ammonium adducts, phosphatidylcholine as an acetate adduct, and cardiolipin, phosphatidate, phosphatidylethanolamine, phosphatidylglycerol, phosphatidylinositol, and phosphatidylserine as deprotonated anions. MS only was used to monitor lysophosphatidate, lysophosphatidylethanolamine, lyso-phosphatidylinositol, lysophosphatidylserine, inositolphosphorylceramide, mannosylinositol-phosphorylceramide, mannosyl-di-(inositolphosphoryl)ceramide as deprotonated anions, and ceramide and lyso-phosphatidylcholine as acetate adducts.

Data were analyzed by Lipotype GmbH with in-house developed lipid identification software based on LipidXplorer (Herzog et al., 2012, Herzog et al., 2011). Data post-processing and normalization were performed by Lipotype GmbH using an in-house developed data management system. Only lipid identifications with a signal-to-noise ratio >5 and a signal intensity fivefold higher than in corresponding blank samples were considered for further data analysis.

The relative abundance of each lipid class was determined by dividing the total amount of lipid species in moles identified in each class by the total amount of lipids identified in the sample in moles. Two-tailed Welch's $t$ test was used to test for differences between WT and $\Delta get3$ samples in each growth medium.

## Reverse transcription quantitative PCR
For extraction of total RNA, yeast cells grown exponentially were lysed by vortexing with glass beads in the presence of equal volumes of acidic phenol pH 4.5–5.0 (Roth) and GTC mix (2 M guanidine thiocyanate, 25 mM Tris-HCl pH 8.0, 5 mM EDTA, 1% (v/v) N-lauroylsarcosine, 150 mM β-mercaptoethanol) for 5 min at 4°C. Extracts were incubated at 65°C for 5 min and then on ice for 5 min before the addition of 1/2 volume of chloroform and 20 mM sodium acetate, 0.2 mM EDTA, and 2 mM Tris-HCl pH 8.0. After centrifugation, the upper aqueous phase was transferred to a fresh tube and an equal volume of phenol/chloroform/isoamylalcohol (25:24:1) was added. Samples were vortexed and centrifuged, and the upper aqueous phase was mixed with an equal volume of chloroform. After vortexing and centrifuging, RNA in the upper phase was precipitated by the addition of 300 mM sodium acetate pH 5.2 and 3 volumes of ethanol. Following centrifugation, the RNA pellet was washed with 70% ethanol and resuspended in RNase-free water.

RNA was quantified and quality-controlled with the $A_{260}/A_{280}$ ratio. DNase treatment was performed on 10 µg total RNA using 2 U TURBO DNase (Thermo Fisher Scientific) in the presence of 40 U RiboLock RNase inhibitor (Thermo Fisher Scientific) for 15 min at 37°C. DNase-treated RNA was purified using the RNA clean and concentrator kit (Zymo Research) according to the manufacturer's instructions.

cDNA was synthesized from 2 µg total RNA in 20 µl total volume in the presence of 2.5 µM oligo-dT primer (T24VN), 0.5 mM dNTP mix, 5 mM DTT, 40 U RiboLock RNase inhibitor (Thermo Fisher Scientific), 200 U Superscript III reverse transcriptase (Thermo Fisher Scientific) and 1× Superscript III reverse transcriptase first strand buffer (Thermo Fisher Scientific). After incubation at 50°C for 1 h, the enzyme was deactivated at 70°C for 15 min. Equal amounts of cDNA were used for qPCR experiments.

Primer pairs were designed to amplify an amplicon of ~130–150 bases. A product melting curve was used to test the amplification of a single amplicon for each primer pair. The range of linearity of cDNA amplification was confirmed, and all used primer pairs had an amplification efficiency of >90%. qPCR was performed using LightCycler 480 SYBR Green I Master (Roche), 0.3 µM primers, and 1× ROX (Reference dye for qPCR; Sigma-Aldrich) in a Mx3000P qPCR machine (Agilent Technologies). Cycling conditions were 95°C for 5 min, 40 cycles of 95°C for 30 s, 55°C for 30 s, and 72°C for 30 s, followed by 95°C for 1 min, 55°C for 30 s and 95°C for 30 s. qPCR data were analyzed using MxPro software. Technical triplicate reactions were performed and values differing from the mean by >0.5 Ct were excluded. Obtained values were normalized to those of *TAF10* and converted using the formula $2^{Ct}$. Finally, converted values were normalized to the average of untreated WT samples.

## Other bioinformatic methods
SQLE structure data file with PDB accession no. 6C6P was retrieved from RCSB PDB (www.rcsb.org) and visualized using VMD (https://www.ks.uiuc.edu/Research/vmd/; Humphrey et al., 1996).

Yeast Erg1 (Uniprot ID: P32476-1) and human SQLE (Uniprot ID: Q14534-1) sequences were retrieved from Uniprot (www.uniprot.org), aligned with Clustal Omega (Sievers et al., 2011) and visualized using Jalview (Waterhouse et al., 2009).

To identify potential hairpin proteins in the SWAT library, proteins with a detectable non-cytosolic signal were selected and queried using the TopologYeast website (http://www.weizmann.ac.il/molgen/TopologYeast/home; Weill et al., 2019). Proteins were considered to contain a potential hairpin if the consensus of the different prediction algorithms predicted exactly two transmembrane segments not more than ten amino acids apart.

## Statistical analysis
Mean, standard deviation, and statistical significance derived from Welch's unequal variances $t$ test (two-tailed, unpaired) were calculated using Excel following standard procedures. Data distribution was assumed to be normal, but this was not formally tested. Differences between samples with P < 0.05 were considered statistically significant.

## Online supplemental material
Fig. S1 contains the lipidomic data complementary to Fig. 1 B. Fig. S2 contains fluorescence microscopy images of GFP-tagged proteins involved in sterol metabolism complementary to Fig. 1 D. Fig. S3 contains the mass spectrometry data complementary to Fig. 2 A. Fig. S4 contains extended information on the hairpin of Erg1 and its role in the protein's function complementary to Fig. 3. Fig. S5 contains fluorescence microscopy images of GFP-tagged proteins predicted to contain a single hairpin complementary to Fig. 7 A. Table S1 contains the background data to the lipidomic analysis presented in Figs. 1 B and S1. Table S2 contains the background data to the mass spectrometry analysis presented in Figs. 2 A and S3. Table S3 contains the list of all yeast proteins in the SWAT library predicted to contain a single hairpin. Table S4 contains the list of all oligos and yeast strains used in this study.

## Data availability
The mass spectrometry proteomics data have been deposited to the ProteomeXchange Consortium via the PRIDE partner repository (https://www.ebi.ac.uk/pride/) with the dataset identifier PXD027705.

All microscopy data associated with this study is accessible on the Göttingen Research Online (GRO) database via the following link: https://doi.org/10.25625/5YTTIQ.

## Acknowledgments

We thank Anne Clancy and Jutta Metz for plasmid construction, Wilhelm Voth for yeast strain production, Bernhard Dobberstein for the anti-Ysy6 antiserum, Christian Klose from Lipotype for advice on sample preparation, Uwe Plessmann for help in MS analysis, and Nica Borgese, Maya Schuldiner, Ursula Jakob, and Kathrin Ulrich for interesting and helpful discussions on this project.

This work was funded by Deutsche Forschungsgemeinschaft SFB 1190 P04 (to K.E. Bohnsack and B. Schwappach) and Z02 (to H. Urlaub), and through Germany's Excellence Strategy - EXC 2067/1-390729940 (to B. Schwappach).

The authors declare no competing financial interests.

Author contributions: Conceptualization: A. Farkas, K.E. Bohnsack, B. Schwappach. Methodology: A. Farkas, H. Urlaub, K.E. Bohnsack, B. Schwappach. Investigation: A. Farkas, H. Urlaub, K.E. Bohnsack. Visualization: A. Farkas. Funding acquisition: H. Urlaub, K.E. Bohnsack, B. Schwappach. Project administration: H. Urlaub, K.E. Bohnsack, B. Schwappach. Supervision: B. Schwappach. Writing – original draft: A. Farkas, B. Schwappach. Writing – review & editing: A. Farkas, H. Urlaub, K.E. Bohnsack, B. Schwappach.

Submitted: 7 January 2022

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

**Supplemental material**

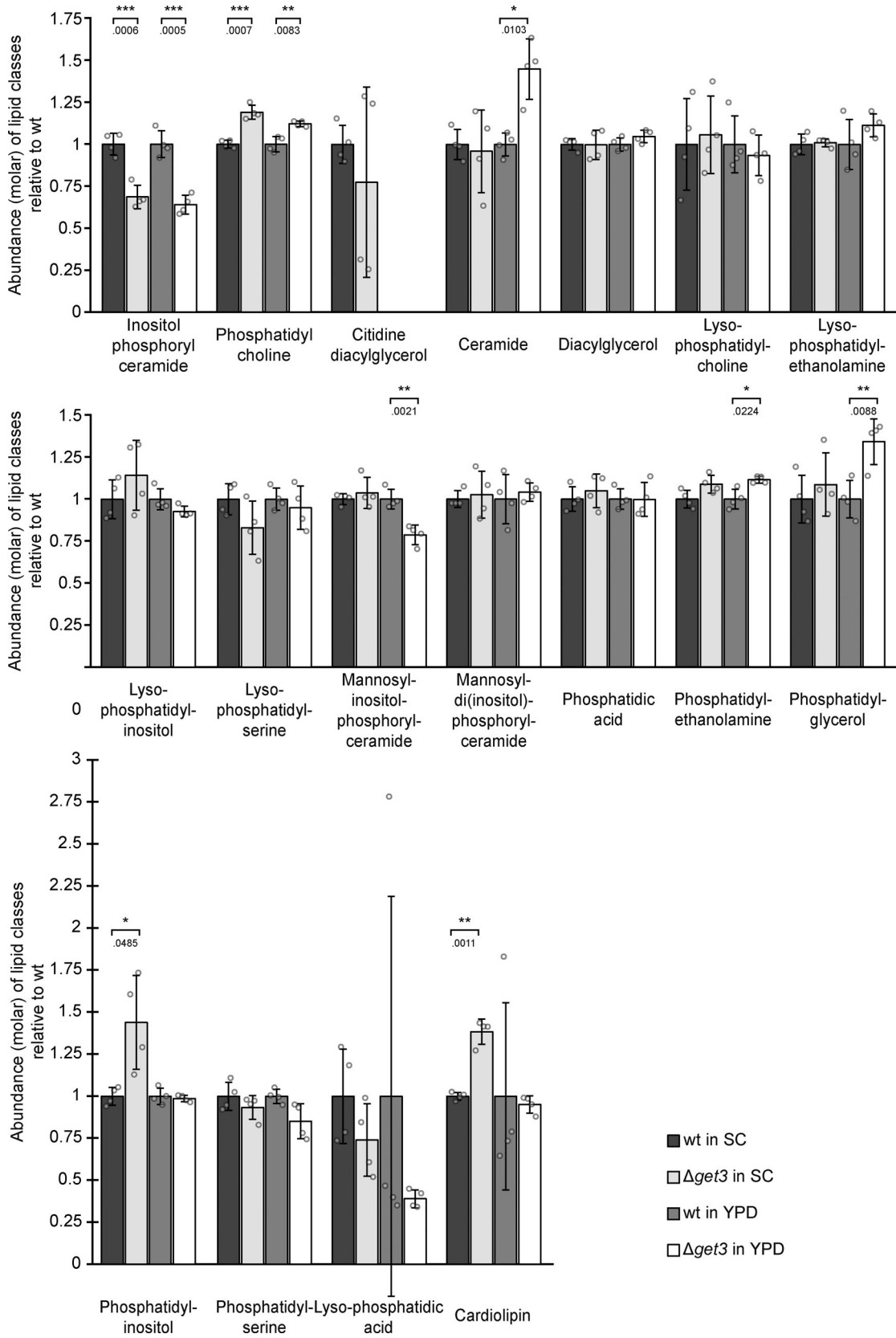

Figure S1. **Lipidomic analysis of WT and Δ*get3* cells.** Lipidomic analysis of WT and Δ*get3* cells in both synthetic complete (SC) and full (yeast extract-peptone-dextrose; YPD) media. Bars represent the average molar abundance of the indicated lipid classes from four biological replicates, with individual data points shown as gray dots, normalized to the WT strain in the respective media. Error bars indicate standard deviation of the mean. The P values calculated using the two-sided Welch's *t* test are represented by stars as follows: * < 0.05; ** < 0.01.

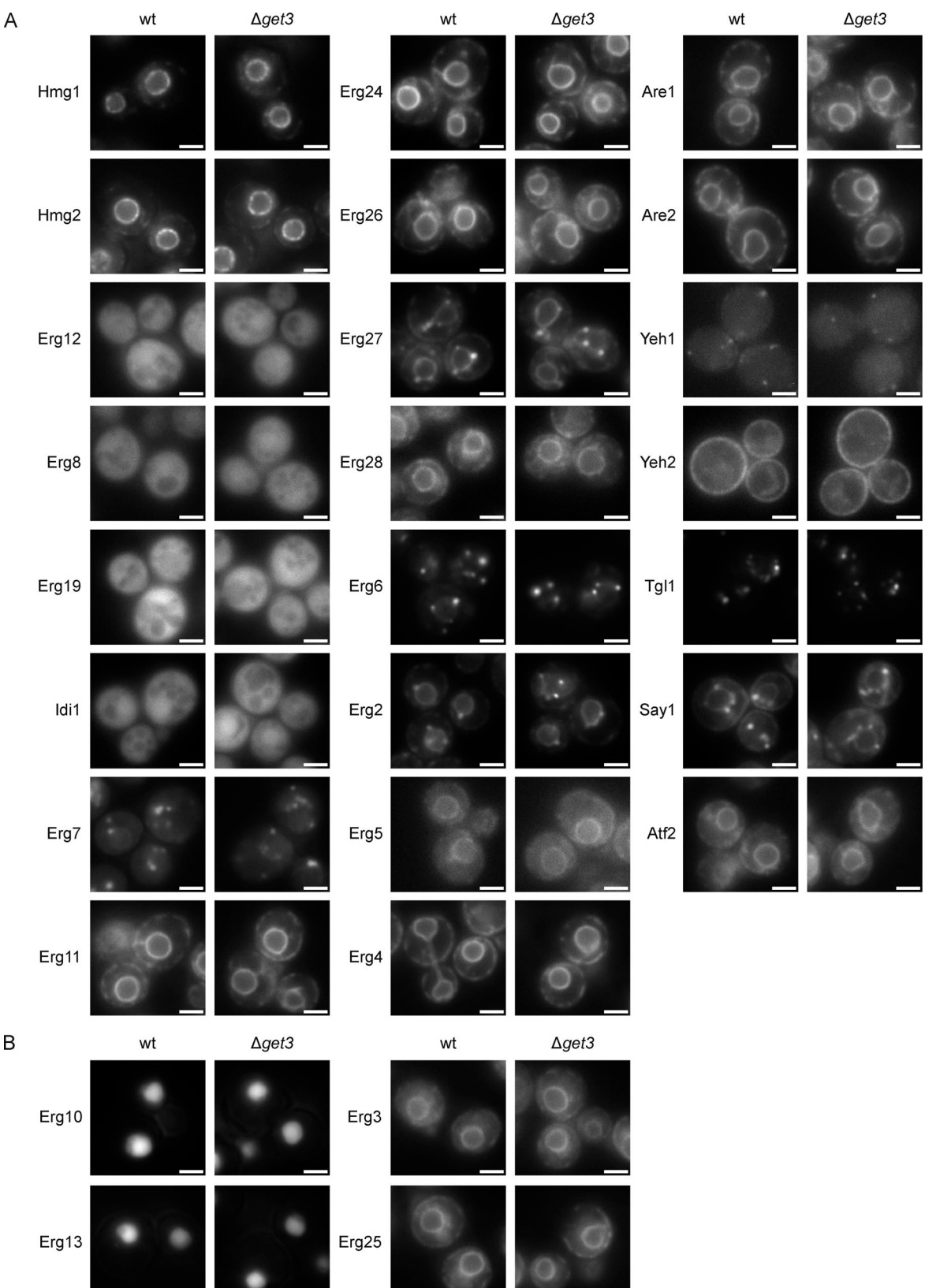

Figure S2. **Fluorescence microscopy images of GFP-tagged proteins involved in sterol metabolism in WT and Δ*get3* cells. (A)** Images of WT and Δ*get3* cells with the indicated proteins tagged N-terminally with GFP, expressed from the *NOP1* promoter. Images are representative of three biological replicates with >100 cells imaged for each replicate. Scale bar: 2 μM. **(B)** Images of WT and Δ*get3* cells with the indicated proteins tagged C-terminally with GFP, expressed from their endogenous promoters. Images are representative of three biological replicates with >100 cells imaged for each replicate. Scale bar: 2 μM.

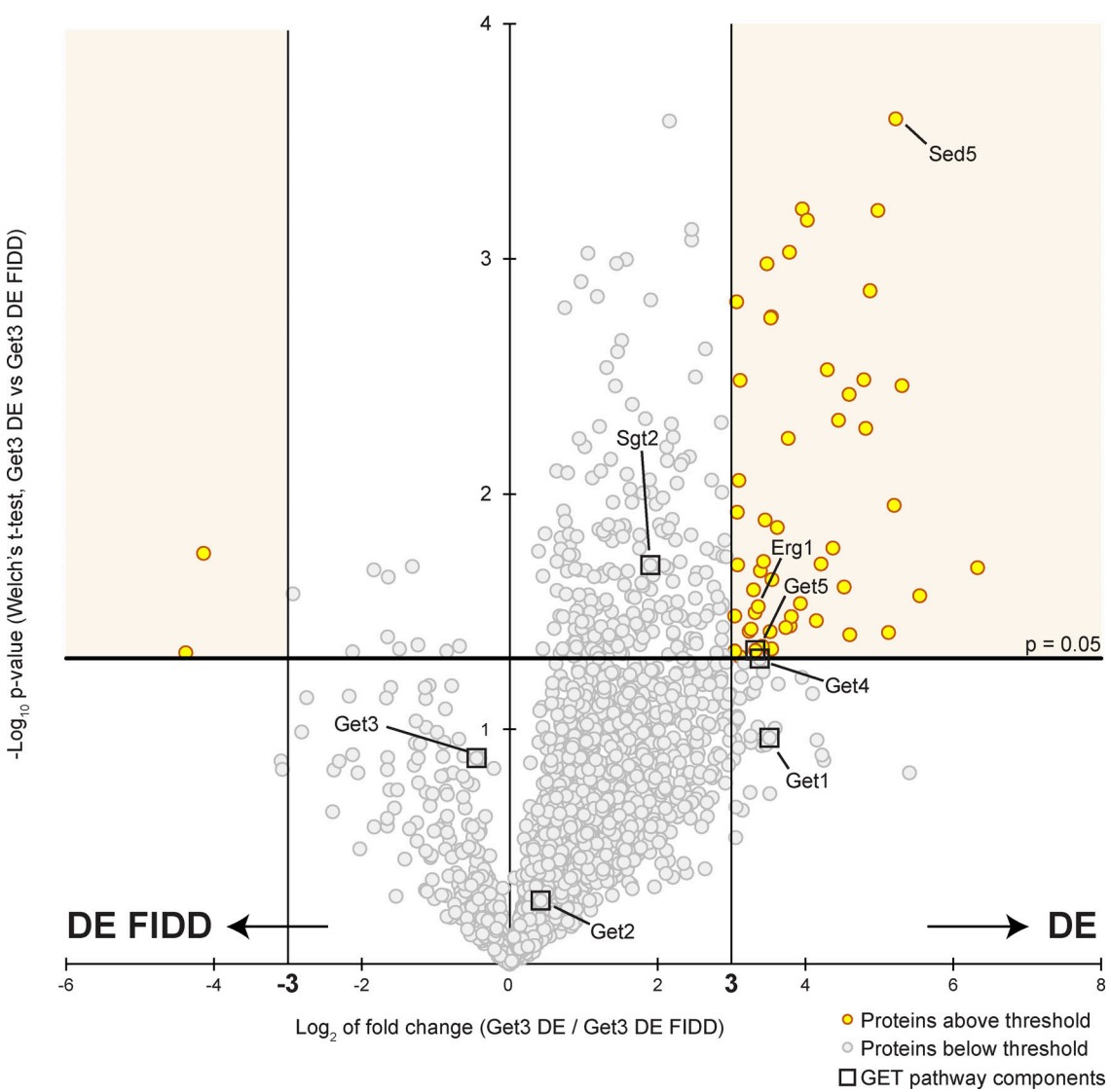

Figure S3. **Analysis of Get3 DE-TEV-GFP and Get3 DE FIDD-TEV-GFP immunoprecipitates by mass spectrometry.** Immunoprecipitation was done in the absence of detergents. The horizontal axis represents the difference of the average $\log_2$ intensity of identified proteins in three biological replicates between Get3 DE and Get3 DE FIDD. The vertical axis represents the $-\log_{10}$ of the P value of the difference between Get3 DE and Get3 DE FIDD for each identified protein calculated with the two-sided Welch's $t$ test. Proteins with a greater than eightfold enrichment and a statistical significance $P < 0.05$ are marked in yellow and GET pathway components are indicated with boxes.

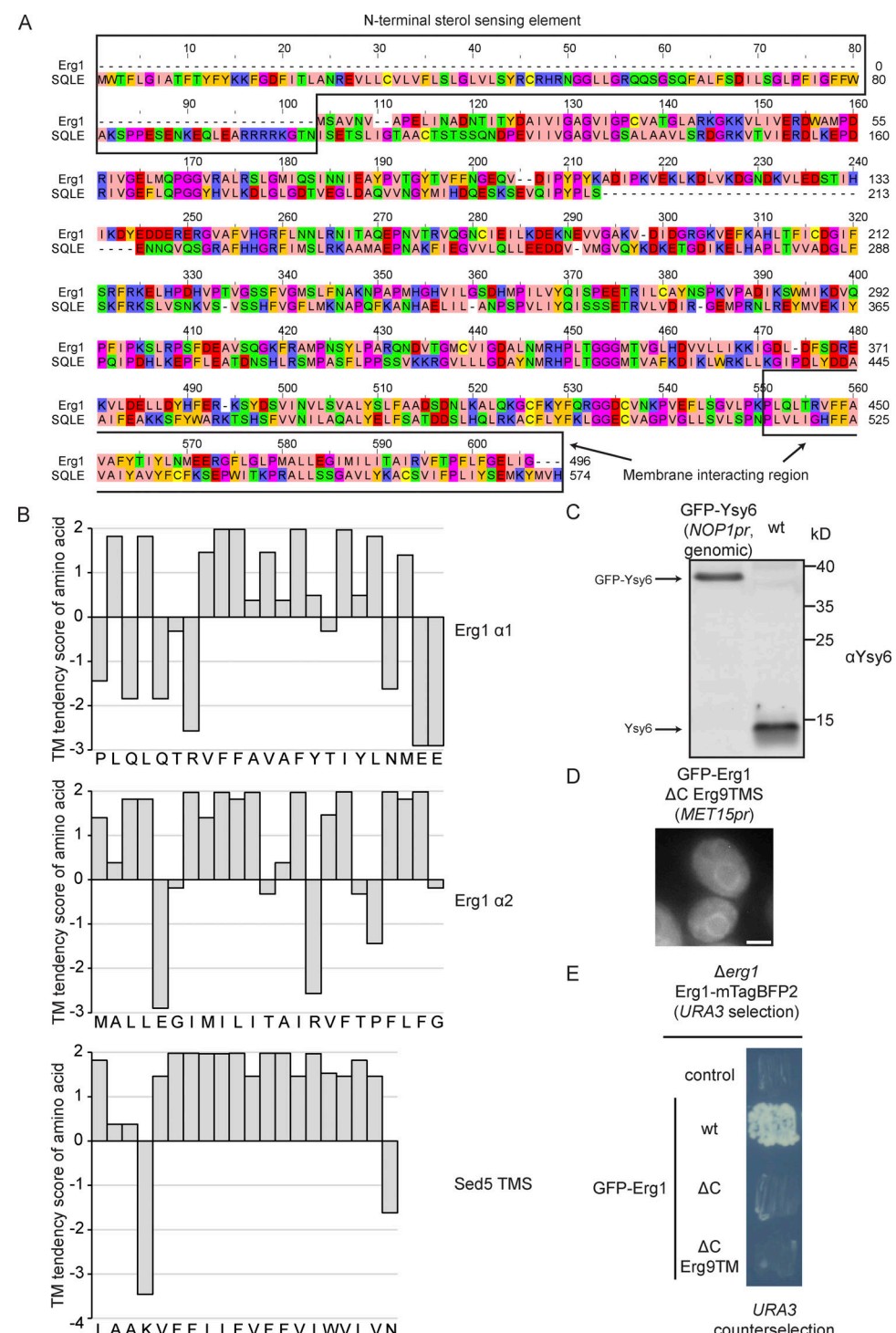

Figure S4.  **Analysis of the properties of the hairpin of Erg1 and its role in the protein's function. (A)** Alignment of the amino acid sequence of yeast Erg1 and human SQLE. Amino acid position in each sequence is indicated at the end of each line. Amino acids are colored according to the "Zappo" scheme in Jalview, i.e., ILVAM is peach, FWY is orange, KRH is blue, DE is red, STNQ is green, PG is purple, C is yellow. Major structural elements referred to in the text are highlighted. **(B)** Hydrophobicity plot of the hairpin helices of Erg1 and the TMS of Sed5 using the TM tendency scores. **(C)** Immunoblot of WT cells or cells expressing N-terminally GFP-tagged Ysy6 from the endogenous locus under control of the *NOP1* promoter detected with the Ysy6 antibody. Image is representative of one biological replicate. **(D)** Fluorescence microscopy image of a WT strain ectopically expressing an N-terminally GFP-tagged Erg1 construct, in which the C-terminal hairpin was substituted with the TMS of Erg9. The image is representative of three biological replicates with >100 cells imaged for each replicate. Scale bar: 2 μM. **(E)** 5-FOA based complementation assay with Erg1 constructs. A Δerg1 strain carrying a *URA3* marker-containing plasmid for expression of Erg1-mTagBFP2 was transformed with plasmids expressing N-terminally GFP-tagged full length Erg1 (WT), Erg1 lacking its C-terminal hairpin (ΔC) or Erg1 with its C-terminal hairpin substituted with the TMS of Erg9. Individual colonies were streaked out onto a synthetic dropout plate containing 5-FOA to drive counterselection of the Erg1-mTagBFP2 plasmid. The image is representative of six colonies streaked out for each plasmid. Source data are available for this figure: SourceData FS4.

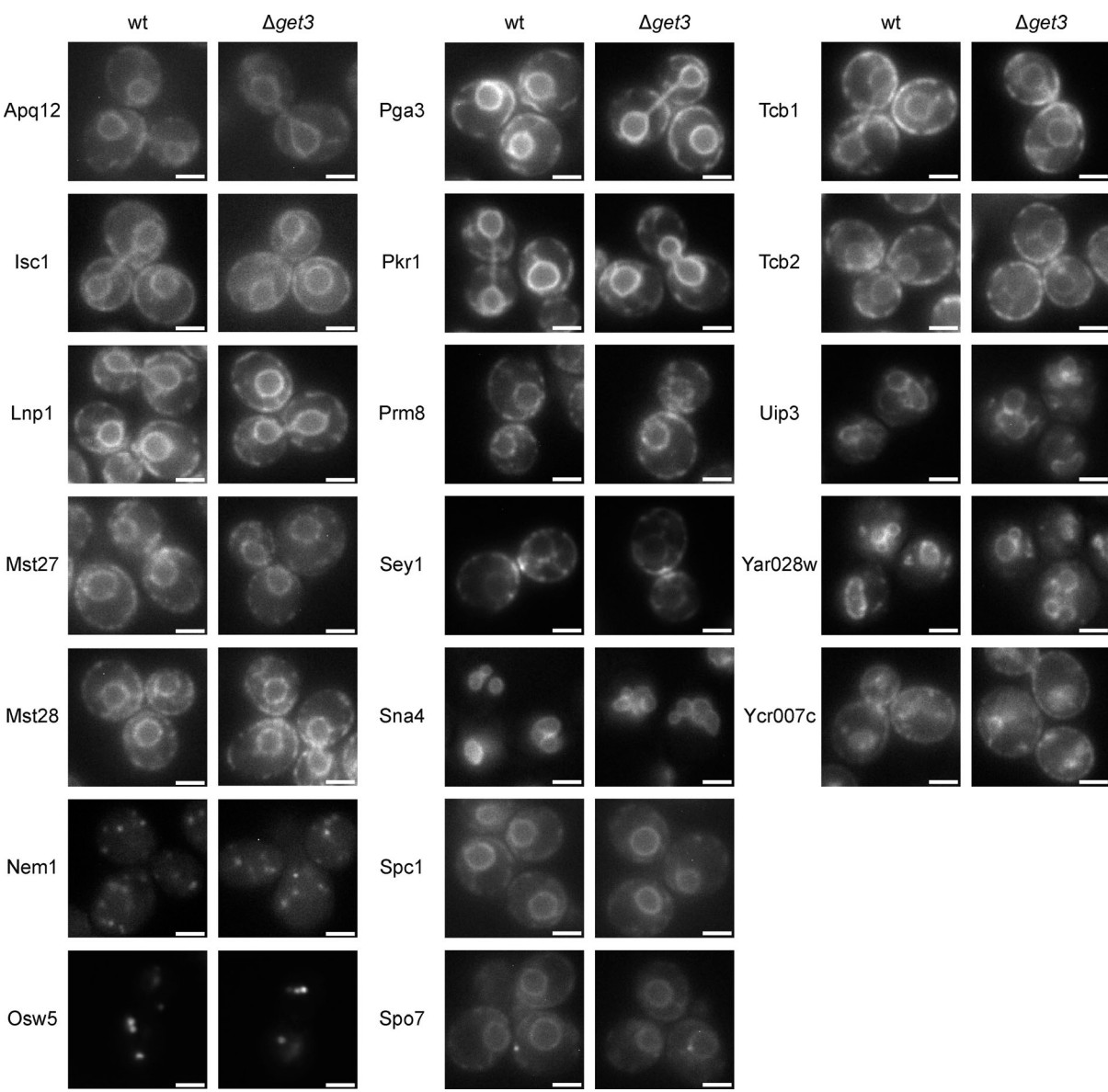

**Figure S5. Fluorescence microscopy images of GFP-tagged proteins predicted to contain a single hairpin in WT and Δget3 cells.** Images of WT and Δget3 cells with the indicated proteins tagged N-terminally with GFP, expressed from the *NOP1* promoter. Images are representative of three biological replicates with >100 cells imaged for each replicate. Scale bar: 2 µM.

**Provided online are Table S1, Table S2, Table S3, and Table S4. Table S1 list Lipidomic analysis of WT and Δget3 strains grown in SC and YPD media. Table S2 list Mass spectrometry analysis of Get3-TEV-GFP DE and Get3-TEV-GFP DE FIDD immunoprecipitates. Table S3 list Yeast proteins in the SWAT library predicted to contain a single hairpin. Table S4 list of plasmids, yeast strains, and oligos used in this study.**

