## [Peer Review File · The Journal of Cell Biology]

Regulated targeting of the monotopic hairpin membrane protein Erg1 requires the GET pathway

Ákos Farkas, Henning Urlaub, Katherine Bohnsack, and Blanche Schwappach

Corresponding Author(s): Blanche Schwappach, Universitätsmedizin Göttingen and Katherine Bohnsack, University Medical Center Goettingen

Review Timeline:

Submission Date:	2022-01-07
Editorial Decision:	2022-02-05
Revision Received:	2022-03-27
Editorial Decision:	2022-03-28
Revision Received:	2022-04-04

Monitoring Editor: William Prinz

Scientific Editor: Dan Simon

Transaction Report:

DOI: <https://doi.org/10.1083/jcb.202201036>

February 5, 2022

Re: JCB manuscript #202201036

Prof. Blanche Schwappach
Universitätsmedizin Göttingen
Molecular Biology
University Medical Center Göttingen Humboldtallee 23
Göttingen D-37073
Germany

Dear Prof. Schwappach,

Thank you for submitting your manuscript entitled "Regulated targeting of the monotopic hairpin protein squalene monooxygenase requires the GET pathway." The manuscript was assessed by expert reviewers, whose comments are appended to this letter. We invite you to submit a revision if you can address the reviewers' key concerns, as outlined here.

You will see that all three reviewers are enthusiastic about your study. Reviewers 2 and 3 have relatively minor concerns, which should all be addressed. The issues raised by Reviewer 1 are more substantial. This reviewer asks you to confirm that Erg1 is not a canonical TA substrate(#1), test direct binding of Get3 to the Erg1 hydrophobic C-tail (#2), and has several minor concerns. All should be addressed. However, it is not necessary to determine whether Get3 interacts with other putative substrates (major point #3), though we will welcome data addressing this issue if you choose to include it.

GENERAL GUIDELINES:

Text limits: Character count for an Article is < 40,000, not including spaces. Count includes title page, abstract, introduction, results, discussion, acknowledgments, and figure legends. Count does not include materials and methods, references, tables, or supplemental legends.

Figures: Articles may have up to 10 main text figures. Figures must be prepared according to the policies outlined in our Instructions to Authors, under Data Presentation, <https://jcb.rupress.org/site/misc/ifora.xhtml>. All figures in accepted manuscripts will be screened prior to publication.

Supplemental information: There are strict limits on the allowable amount of supplemental data. Articles may have up to 5 supplemental figures. Up to 10 supplemental videos or flash animations are allowed. A summary of all supplemental material should appear at the end of the Materials and methods section.

Please note that JCB now requires authors to submit Source Data used to generate figures containing gels and Western blots with all revised manuscripts. This Source Data consists of fully uncropped and unprocessed images for each gel/blot displayed in the main and supplemental figures. Since your paper includes cropped gel and/or blot images, please be sure to provide one Source Data file for each figure that contains gels and/or blots along with your revised manuscript files. File names for Source Data figures should be alphanumeric without any spaces or special characters (i.e., SourceDataF#, where F# refers to the associated main figure number or SourceDataFS# for those associated with Supplementary figures). The lanes of the gels/blots should be labeled as they are in the associated figure, the place where cropping was applied should be marked (with a box), and molecular weight/size standards should be labeled wherever possible.

As you may know, the typical timeframe for revisions is three to four months. However, we at JCB realize that the implementation of measures that limit spread of COVID-19 also pose challenges to scientific researchers. Therefore, JCB has

waived the revision time limit. If necessary, we recommend that you reach out to the editors to decide on an appropriate time frame for resubmission. Please note that papers are generally considered through only one revision cycle, so any revised manuscript will likely be either accepted or rejected.

Thank you for this interesting contribution to Journal of Cell Biology. You can contact us at the journal office with any questions, cellbio@rockefeller.edu or call (212) 327-8588.

Sincerely,

William Prinz, PhD
Monitoring Editor
Journal of Cell Biology

Dan Simon, PhD
Scientific Editor
Journal of Cell Biology

Reviewer #1 (Comments to the Authors (Required)):

In this manuscript, Farkas et al., provide new evidence that hairpin proteins use the GET pathway for targeting to the ER membrane. It is well established that the central Get3 ATPase in the GET pathway captures C-terminally anchored transmembrane proteins (TA proteins) and targets them to the ER membrane for insertion by the Get1/2 membrane insertase. The authors observed that Δ Get3 yeast strain is sensitive to the antifungal drug terbinafine, suggesting that sterol synthesis is impaired in Δ Get3 strain. Subsequently, the authors show that Erg1, an enzyme involved in sterol synthesis, is not efficiently localized to the ER and is mislocalized as cytosolic aggregates in Δ Get3 cells. Unlike typical Get3 clients, Erg1 does not have an obvious C-terminal TMD and instead, it contains a C-terminal long hydrophobic sequence of about 50 amino acids, which was predicted to be a hairpin. Using a Get3 mutant (FIDD) that is defective in binding to the TMD, the authors show that Erg1 likely binds to the TMD binding groove of Get3. The authors further went on to identify more hairpin proteins that could potentially use the GET pathway. Overall, this manuscript is well organized with high-quality data. However, the manuscript will be strengthened if the authors provide sufficient evidence that Erg1 is not a TA protein, and address how Get3 binds and shields the hydrophobic hairpin sequence.

Major concerns:

1. The authors need to experimentally show that Erg1 is not a canonical TA substrate since Uniprot predicts Erg1 contains a C-terminal TMD. The authors can do this by adding an N-glycosylation motif to the C-terminus of Erg1 in comparison to a control substrate Sed5. If possible, they can check if Erg1 is dissociated from the ER membrane by sodium carbonate extraction to determine if Erg1 is peripherally associated with the ER membrane as a monotypic protein.
2. The authors need to determine how Get3 shields the long hydrophobic C-terminal sequence (~50 amino acids) of Erg1. Within the 50 amino acids sequence, it contains a small hydrophobic region and a large hydrophobic region at the very C-terminus that resembles the TMD of a TA protein. The authors can delete region 1 and region 2 individually within the C-terminal 50 amino acids of Erg1 and test the interaction with Get3 as done in Figure 3D. This will hopefully establish whether Get3 binds to both regions or binds to either one.
3. The authors should test the interaction of Get3 (DE) or Get3 (DE/FIDD) with new substrates (Lam1, Sip3, Prm9, Tsc10, and Ubx2) shown in Figure 7. This data will exclude the possibility that mislocalization of these substrates is caused by an indirect effect in Δ Get3 cells.

Minor concerns:

1. It would be helpful if Get3 deleted strain is labeled as Δ Get3 because labeling it as Get3 is confusing as to if Get3 is expressed or deleted in this strain.
2. Why Get1/2 deleted show a strong phenotype compared to Get3 deleted strain in Figure 2D. Is it possible that Get1/2 can partially mediate the insertion of these proteins even in the absence of Get3?

3. It will be helpful to include a supplemental figure comparing hydrophobicity plots of Erg1 hairpin and a canonical TA protein.

4. I feel that Doa10 results are not very relevant to the current manuscript.

Reviewer #2 (Comments to the Authors (Required)):

This manuscript by Farkas and colleagues describes that the GET pathway, aside a well-characterized role in the biogenesis of tail-anchored proteins, also contributes to the targeting of some hairpin-containing proteins. In particular, it is shown that the sensitivity of Get3 mutants to terbinafine, an inhibitor of the yeast squalene monooxygenase Erg1, results from defects in Erg1 localization. Biochemical data support a direct role of Get3 in Erg1 membrane targeting by binding via its hydrophobic substrate-binding groove, to erg1 C-terminal hydrophobic domain. This Get3 function also appears to be conserved for human SQLE, at least when expressed in yeast. Intriguingly, microscopy data suggest that other hairpin-containing proteins also rely on Get3 to target to the ER.

Overall, the manuscript presents good quality and novel data on the role of the GET pathway in controlling ER protein targeting. This new message will certainly be of interest to the cell biology community, in particular to those with interest in membrane biology. Below are a few points that if addressed would clarify and strengthen some of the conclusions of this interesting study.

- I wonder if targeting Erg1 to the ER membrane independently of the GET pathway (for example by fusing Erg1 to the tail anchor of Erg9, an EMC client) rescues Terbinafine sensitivity of Get3 mutants.

- Several hairpin-containing proteins show localization defects in Get3 mutants. Presumably due to their low expression levels none of these was picked up in the substrate-trapping experiment (Figure 2). I wonder if among the protein enriched in Get3 precipitates (over the FIDD mutant) are any monotopic/hairpin proteins aside from Erg1?

- There is now substantial information on the mechanisms by which TA proteins bind to Get3, are delivered to Get1/2 and membrane inserted by this complex. How does the authors envision these steps to occur in the case of hairpin-containing proteins? In my opinion these points should be discussed even if a definitive answer to these questions isn't available at the moment.

Reviewer #3 (Comments to the Authors (Required)):

This is an excellent paper that convincingly shows a novel role for the conserved GET protein complex in insertion of hairpin transmembrane domains into the ER. Much of the work is focused on the sterol biosynthetic enzyme Erg1, showing that this protein is inserted into the ER by GET and a physiological role for this in response to sterol deprivation. The authors show that this pathway has wider implications in biology given that GET affects the ER insertion of other proteins with hairpin transmembrane domains. They also show their findings are evolutionary conserved, as human Erg1 (SQLE) expressed in yeast also is inserted into the ER by GET. All major claims are well supported by direct experimental evidence.

Minor suggestions:

1. Why not test the sensitivity of Erg1 Δ C to terbinafine to prove that its insertion into the ER by GET is required for function?
2. Quantification of images in Fig 4A would be helpful to compare ER insertion with total protein levels by western blot.

We thank the referees for their feedback and constructive comments to our work. Below, we provide a complete point-by-point response to the reviewers' specific suggestions. The corresponding modifications to the text are indicated in yellow in the word file of the manuscript.

In brief, we have added the following data to the manuscript:

Figure 3D – Western blot analysis of glycosylation of HA-Erg1, Ysy6 and Sed5 carrying wild-type and non-glycosylatable opsin tags.

Figure 3E – Analysis of extraction of HA-Erg1, Get3 and Sed5 from microsomes by different treatments.

Figure 3H – Analysis of interaction between Get3 and Erg1 lacking individual elements of its hairpin ($\alpha 1$ and $\alpha 2$) by immunoprecipitation.

Figure 3I – Quantification of Figure 3H.

Figure 4B – Quantification of relative ER and cytosolic signal in microscopy images.

Figure S4B – Hydrophobicity plots of Erg1 $\alpha 1$, Erg1 $\alpha 2$ and Sed5 TMS

Figure S4C – Fluorescence microscopy analysis of Erg1 Δ C conjugated to TMS of Erg9.

Figure S4D – 5FOA assay analysing ability of Erg1 derivatives to functionally complement Δ erg1.

Reviewer 1

In this manuscript, Farkas et al., provide new evidence that hairpin proteins use the GET pathway for targeting to the ER membrane. It is well established that the central Get3 ATPase in the GET pathway captures C-terminally anchored transmembrane proteins (TA proteins) and targets them to the ER membrane for insertion by the Get1/2 membrane insertase. The authors observed that Δ Get3 yeast strain is sensitive to the antifungal drug terbinafine, suggesting that sterol synthesis is impaired in Δ Get3 strain. Subsequently, the authors show that Erg1, an enzyme involved in sterol synthesis, is not efficiently localized to the ER and is mislocalized as cytosolic aggregates in Δ Get3 cells. Unlike typical Get3 clients, Erg1 does not have an obvious C-terminal TMD and instead, it contains a C-terminal long hydrophobic sequence of about 50 amino acids, which was predicted to be a hairpin. Using a Get3 mutant (FIDD) that is defective in binding to the TMD, the authors show that Erg1 likely binds to the TMD binding groove of Get3. The authors further went on to identify more hairpin proteins that could potentially use the GET pathway. Overall, this manuscript is well organized with high-quality data. However, the manuscript will be strengthened if the authors provide sufficient evidence that Erg1 is not a TA protein, and address how Get3 binds and shields the hydrophobic hairpin sequence.

Major concerns:

1. Point: The authors need to experimentally show that Erg1 is not a canonical TA substrate since Uniprot predicts Erg1 contains a C-terminal TMD. The authors can do this by adding an N-glycosylation motif to the C-terminus of Erg1 in comparison to a control substrate Sed5. If possible, they can check if Erg1 is dissociated from the ER membrane by sodium carbonate extraction to determine if Erg1 is peripherally associated with the ER membrane as a monotopic protein.

Response: We have generated constructs for the expression of Erg1 with a C-terminal opsin-tag containing an N-glycosylation site and a non-glycosylatable derivative in which the glycosylated arginine is substituted with glutamine (N/Q) to allow unambiguous identification of glycosylated proteins. As additional controls, analogous constructs for the expression of the TA proteins Sed5 and Ysy6 were also generated. Glycosylation retards protein migration in polyacrylamide gels and while substantial and minimal portions of Ysy6 and Sed5 respectively were observed as a slower migrating species, no specific signal corresponding to glycosylated Erg1 was detectable. These results, shown in Fig. 3D, imply that the C-terminus of Erg1 does not reach the ER lumen. Furthermore, we performed protein extraction with different buffers containing sodium carbonate or detergents. This revealed that, similar to the monotopic hairpin protein Tsc10, Erg1 is partially released from the membrane by treatment with high salt, protein denaturants, or sodium carbonate and is only fully extracted upon exposure to detergents. This result, shown in Fig. 3E, further supports the model that Erg1 assumes a monotopic topology. Alongside describing these results, we have adjusted the text to emphasize that Erg1 likely behaves as a monotopic integral membrane protein as has been demonstrated before for other hairpin proteins in yeast (Gupta et al., 2009).

2. Point: The authors need to determine how Get3 shields the long hydrophobic C-terminal sequence (~50 amino acids) of Erg1. Within the 50 amino acids sequence, it contains a small hydrophobic region and a large hydrophobic region at the very C-terminus that resembles the TMD of a TA protein. The authors can delete region 1 and region 2 individually within the C-terminal 50 amino acids of Erg1 and test the interaction with Get3 as done in Figure 3D. This will hopefully establish whether Get3 binds to both regions or binds to either one.

Response: Constructs for the expression of Erg1 lacking either of the two hydrophobic regions (termed $\alpha 1$ and $\alpha 2$) were generated to dissect the contributions that each of these regions makes to the interaction with Erg1. Immunoprecipitation experiments followed by Western blotting revealed that lack of either of the hydrophobic elements impairs Get3 interaction but that lack of the $\alpha 2$ region, which is more hydrophobic, more strongly reduces

the amount of co-precipitated Get3. These data are shown in Fig. 3F-G and are described in the revised manuscript.

3. Point: The authors should test the interaction of Get3 (DE) or Get3 (DE/FIDD) with new substrates (Lam1, Sip3, Prm9, Tsc10, and Ubx2) shown in Figure 7. This data will exclude the possibility that mislocalization of these substrates is caused by an indirect effect in Δ Get3 cells.

Response: The mis-localization of Lam1, Sip3, Tsc10, and Ubx2 in cells lacking Get3 and the exacerbation of these phenotypes upon overexpression of these hairpin proteins highlights them as potential Get3 clients. While demonstrating physical association with Get3 would further corroborate this notion, demonstrating Get3-substrate interactions via co-immunoprecipitation experiments is notoriously challenging. Indeed, our mass spectrometry experiment only recovered Sed5 as a clearly trapped TA substrate of Get3, and analogous experiments in mammalian cells uncovered only a few TA substrates as well (see Coy-Vergara et al., 2019). We assume that the lack of more co-precipitating substrates in such experiments is the result of a combination of the low expression level of substrates, stability of the Get3-substrate complex, saturation of Get3 with preferred clients, and that other targeting pathways may take care of a considerable portion of the flux of de-novo synthesized substrate proteins. As the editor indicated that providing such data is not essential for the current manuscript, and considering the time it would take to establish alternative ways to demonstrate a physical interaction of these putative clients with Get3, this was not pursued further. However, in the discussion section of our manuscript, we state that these putative clients were not recovered in our mass spectrometry analysis of immunoprecipitated Get3 interactors and discuss explanations for this.

Minor concerns:

4. Point: It would be helpful if Get3 deleted strain is labeled as Δ Get3 because labeling it as Get3 is confusing as to if Get3 is expressed or deleted in this strain.

Response: Get3 deleted strains are now labelled as Δ get3, and the same nomenclature has been applied to all other deletion strains used in this study as well.

5. Point: Why Get1/2 deleted show a strong phenotype compared to Get3 deleted strain in Figure 2D. Is it possible that Get1/2 can partially mediate the insertion of these proteins even in the absence of Get3?

Response: Analogous to our findings that Erg1 mislocalization and sensitivity to terbinafine are more pronounced in a Δ get1 Δ get2 strain than a Δ get3, it has been previously observed

that loss of the GET receptor creates a stronger phenotype than lack of Get3 (Jonikas et al., 2009). The explanation probably lies in the fact that, in the absence of the GET receptor, Get3 and its substrates accumulate in protein aggregates (Jonikas et al., 2009; Powis et al., 2013; Schuldiner et al., 2008), thus preventing delivery of clients via alternative targeting pathways and causing proteotoxic stress in the cell. Nonetheless, the possibility of alternative cytosolic chaperones delivering clients to the receptor in the absence of Get3 is indeed an interesting avenue that could be explored in the future. We include this hypothesis for the observed effects in the manuscript.

6. Point: It will be helpful to include a supplemental figure comparing hydrophobicity plots of Erg1 hairpin and a canonical TA protein.

Response: In Fig. S4B, we now show hydrophobicity plots of both elements of the Erg1 hairpin ($\alpha 1$ and $\alpha 2$) and, for comparison, the TMS of Sed5. This analysis shows that both $\alpha 1$ and $\alpha 2$ of Erg1 are less hydrophobic than the TMS of Sed5 due to the presence of hydrophilic residues punctuating the hydrophobic stretch. Furthermore, $\alpha 2$ displays a higher hydrophobicity than $\alpha 1$, which is consistent with our finding that deletion of the second helix impairs interaction with Get3 more strongly than lack of the first.

7. Point: I feel that Doa10 results are not very relevant to the current manuscript.

Response: Erg1 is extensively regulated by both, transcription and Doa10-mediated degradation. Our results confirm that part of the reason why the loss of Erg1 from the ER membrane is so striking when Get3-dependent targeting is lost, is that Doa10 actively degrades ER-resident Erg1. Thus, these findings support the notion that the expression and degradation dynamics are important parameters to consider when looking for substrates of targeting pathways, as mentioned in the discussion. Therefore, although we agree that the Doa10 results are not directly related to the biosynthetic targeting of Erg1 to the ER membrane, we feel that presenting these data is important for comprehensively understanding the influence of Get3 on Erg1 and supporting the other conclusions of the study.

Reviewer 2

This manuscript by Farkas and colleagues describes that the GET pathway, aside a well-characterized role in the biogenesis of tail-anchored proteins, also contributes to the targeting of some hairpin-containing proteins. In particular, it is shown that the sensitivity of Get3 mutants to terbinafine, an inhibitor of the yeast squalene monooxygenase Erg1, results from defects in Erg1 localization. Biochemical data support a direct role of Get3 in Erg1 membrane

targeting by binding via its hydrophobic substrate-binding groove, to erg1 C-terminal hydrophobic domain. This Get3 function also appears to be conserved for human SQLE, at least when expressed in yeast. Intriguingly, microscopy data suggest that other hairpin-containing proteins also rely on Get3 to target to the ER.

Overall, the manuscript presents good quality and novel data on the role of the GET pathway in controlling ER protein targeting. This new message will certainly be of interest to the cell biology community, in particular to those with interest in membrane biology. Below are a few points that if addressed would clarify and strengthen some of the conclusions of this interesting study.

1. Point: I wonder if targeting Erg1 to the ER membrane independently of the GET pathway (for example by fusing Erg1 to the tail anchor of Erg9, an EMC client) rescues Terbinafine sensitivity of Get3 mutants.

Response: We have generated construct for the expression of Erg1 in which the hairpin region is replaced by the TA of Erg9. However, as shown in Fig. S4D, expression of this chimeric protein failed to functionally rescue lack of Erg1 in a 5-FOA assay and thus the construct is not suitable to test the rescue of terbinafine sensitivity. However, this experiment allowed us to confirm that the specific arrangement provided by the hairpin is vital for the function of Erg1, which we now highlight in the revised manuscript.

2. Point: Several hairpin-containing proteins show localization defects in Get3 mutants. Presumably due to their low expression levels none of these was picked up in the substrate-trapping experiment (Figure 2). I wonder if among the protein enriched in Get3 precipitates (over the FIDD mutant) are any monotopic/hairpin proteins aside from Erg1?

Response: We did not find other monotopic/hairpin proteins in the Get3 precipitates, however, as discussed in the manuscript, isolating substrates with Get3 is challenging and the approach employed here likely revealed only the most robust clients of Get3, as evidenced by the fact that the only TA protein recovered in the screen in high abundance and with statistical significance was Sed5. Identifying clients of targeting pathways *in vivo* to demonstrate direct physical interaction between substrates and targeting pathways has been challenging and other methods would be likely necessary to detect interactions which occur with a lower affinity and less frequently. A complete list of the proteins identified in our mass spectrometry analyses is presented in Table S2.

3. Point: There is now substantial information on the mechanisms by which TA proteins bind to Get3, are delivered to Get1/2 and membrane inserted by this complex. How does the authors envision these steps to occur in the case of hairpin-containing proteins? In my opinion

these points should be discussed even if a definitive answer to these questions isn't available at the moment.

Response: The targeting process of hairpin protein via the GET pathway is likely initiated by recognition of the hydrophobic helices by the pretargeting complex, followed by transfer to Get3. Our results show that Get3 recognizes both helices of the Erg1 hairpin, although the effect of the deletion of individual helices is not additive, as the loss of the more hydrophobic helix almost completely abolishes the interaction between Erg1 and Get3, whereas loss of the other one does not. This suggests that the hairpin is recognized at least to some extent differently from single hydrophobic helices. Since Get3 and its evolutionary homologs are known to be able to form multimeric complexes capable of substrate binding (Bozkurt et al., 2009; Suloway et al., 2012), it is possible that a multimeric form of Get3 is involved in the binding of helices arranged as a hairpin. To insert into the membrane, the hairpin also needs to bypass the charged surface of the membrane and reach its hydrophobic core. Thus, similarly to TA proteins, the Get1/2 receptor complex may be able to provide a conduit for the hydrophobic helices of hairpin proteins as well once delivered by Get3. This has been incorporated into the discussion as well.

Reviewer 3

This is an excellent paper that convincingly shows a novel role for the conserved GET protein complex in insertion of hairpin transmembrane domains into the ER. Much of the work is focused on the sterol biosynthetic enzyme Erg1, showing that this protein is inserted into the ER by GET and a physiological role for this in response to sterol deprivation. The authors show that this pathway has wider implications in biology given that GET affects the ER insertion of other proteins with hairpin transmembrane domains. They also show their findings are evolutionary conserved, as human Erg1 (SQLE) expressed in yeast also is inserted into the ER by GET. All major claims are well supported by direct experimental evidence.

Minor suggestions:

1. Point: Why not test the sensitivity of Erg1 Δ C to terbinafine to prove that its insertion into the ER by GET is required for function?

Response: Expression of Erg1 Δ C failed to functionally rescue lack of Erg1 as shown in a 5-FOA assay demonstrating that the presence of the hairpin and/or membrane localization is essential for its function. This assay allows us to draw an analogous conclusion to a growth analysis on terbinafine and is present in Fig. S4D of the revised manuscript.

2. Point: Quantification of images in Fig 4A would be helpful to compare ER insertion with total protein levels by western blot.

Response: We have now added a quantification of the images comparing the strength of the ER and lipid droplet signal to the average cellular fluorescence (Fig. 4B), which supports the conclusion of Erg1 mislocalization to the cytosol in the absence of terbinafine and a more pronounced difference between wt and $\Delta get3$ cells in the presence of terbinafine.

March 28, 2022

RE: JCB Manuscript #202201036R

Prof. Blanche Schwappach
Universitätsmedizin Göttingen
Molecular Biology
University Medical Center Göttingen Humboldtallee 23
Göttingen D-37073
Germany

Dear Prof. Schwappach,

Thank you for submitting your revised manuscript entitled "Regulated targeting of the monotopic hairpin protein squalene monooxygenase requires the GET pathway." We would be happy to publish your paper in JCB pending final revisions necessary to meet our formatting guidelines (see details below).

A. MANUSCRIPT ORGANIZATION AND FORMATTING:

- 1) Text limits: Character count for Articles is < 40,000, not including spaces. Count includes title page, abstract, introduction, results, discussion, and acknowledgments. Count does not include materials and methods, figure legends, references, tables, or supplemental legends.
- 2) Figure formatting: Articles may have up to 10 main text figures. Scale bars must be present on all microscopy images, including inset magnifications. Molecular weight or nucleic acid size markers must be included on all gel electrophoresis.
- 3) Statistical analysis: Error bars on graphic representations of numerical data must be clearly described in the figure legend. The number of independent data points (n) represented in a graph must be indicated in the legend. Statistical methods should be explained in full in the materials and methods. For figures presenting pooled data the statistical measure should be defined in the figure legends. Please also be sure to indicate the statistical tests used in each of your experiments (both in the figure legend itself and in a separate methods section) as well as the parameters of the test (for example, if you ran a t-test, please indicate if it was one- or two-sided, etc.). Also, if you used parametric tests, please indicate if the data distribution was tested for normality (and if so, how). If not, you must state something to the effect that "Data distribution was assumed to be normal but this was not formally tested."
- 4) Title: While your current title will be appreciated by the specialists, we do not feel that it will be accessible to a broader cell biology audience. Therefore we suggest the following title: "Regulated targeting of the monotopic hairpin membrane protein Erg1 requires the GET pathway."
- 5) Materials and methods: Should be comprehensive and not simply reference a previous publication for details on how an experiment was performed. Please provide full descriptions (at least in brief) in the text for readers who may not have access to referenced manuscripts. The text should not refer to methods "...as previously described."
- 6) For all cell lines, vectors, constructs/cDNAs, etc. - all genetic material: please include database / vendor ID (e.g., Addgene, ATCC, etc.) or if unavailable, please briefly describe their basic genetic features, even if described in other published work or gifted to you by other investigators (and provide references where appropriate). Please be sure to provide the sequences for all of your oligos: primers, si/shRNA, RNAi, gRNAs, etc. in the materials and methods. You must also indicate in the methods the source, species, and catalog numbers/vendor identifiers (where appropriate) for all of your antibodies, including secondary. If antibodies are not commercial please add a reference citation if possible.
- 7) Microscope image acquisition: The following information must be provided about the acquisition and processing of images:
 - a. Make and model of microscope
 - b. Type, magnification, and numerical aperture of the objective lenses
 - c. Temperature
 - d. Imaging medium
 - e. Fluorochromes

f. Camera make and model

g. Acquisition software

h. Any software used for image processing subsequent to data acquisition. Please include details and types of operations involved (e.g., type of deconvolution, 3D reconstitutions, surface or volume rendering, gamma adjustments, etc.).

8) References: There is no limit to the number of references cited in a manuscript. References should be cited parenthetically in the text by author and year of publication. Abbreviate the names of journals according to PubMed.

9) Supplemental materials: There are strict limits on the allowable amount of supplemental data. Articles/Tools may have up to 5 supplemental figures and 10 videos. Please also note that tables, like figures, should be provided as individual, editable files. A summary of all supplemental material should appear at the end of the Materials and methods section. Please include one brief sentence per item.

10) eTOC summary: A ~40-50 word summary that describes the context and significance of the findings for a general readership should be included on the title page. The statement should be written in the present tense and refer to the work in the third person. It should begin with "First author name(s) et al..." to match our preferred style.

11) Conflict of interest statement: JCB requires inclusion of a statement in the acknowledgements regarding competing financial interests. If no competing financial interests exist, please include the following statement: "The authors declare no competing financial interests." If competing interests are declared, please follow your statement of these competing interests with the following statement: "The authors declare no further competing financial interests."

12) A separate author contribution section is required following the Acknowledgments in all research manuscripts. All authors should be mentioned and designated by their first and middle initials and full surnames. We encourage use of the CRediT nomenclature (<https://casrai.org/credit/>).

13) ORCID IDs: ORCID IDs are unique identifiers allowing researchers to create a record of their various scholarly contributions in a single place. At resubmission of your final files, please consider providing an ORCID ID for as many contributing authors as possible.

14) Please note that JCB now requires authors to submit Source Data used to generate figures containing gels and Western blots with all revised manuscripts. This Source Data consists of fully uncropped and unprocessed images for each gel/blot displayed in the main and supplemental figures. Since your paper includes cropped gel and/or blot images, please be sure to provide one Source Data file for each figure that contains gels and/or blots along with your revised manuscript files. File names for Source Data figures should be alphanumeric without any spaces or special characters (i.e., SourceDataF#, where F# refers to the associated main figure number or SourceDataFS# for those associated with Supplementary figures). The lanes of the gels/blots should be labeled as they are in the associated figure, the place where cropping was applied should be marked (with a box), and molecular weight/size standards should be labeled wherever possible. Source Data files will be directly linked to specific figures in the published article. Source Data Figures should be provided as individual PDF files (one file per figure). Authors should endeavor to retain a minimum resolution of 300 dpi or pixels per inch. Please review our instructions for export from Photoshop, Illustrator, and PowerPoint here: <https://rupress.org/jcb/pages/submission-guidelines#revised>.

B. FINAL FILES:

Thank you for this exciting contribution, we look forward to publishing your paper in Journal of Cell Biology.

Sincerely,

William Prinz, PhD
Monitoring Editor
Journal of Cell Biology

Dan Simon, PhD
Scientific Editor
Journal of Cell Biology